# Portimine A toxin causes skin inflammation through ZAKα-dependent NLRP1 inflammasome activation

Léana Gorse [1,22], Loïc Plessis [2,3,22], Stephen Wearne[4,22], Margaux Paradis [1,22], Miriam Pinilla [5,6,22], Rae Chua[7,22], Seong Soo Lim[4], Elena Pelluz [5,6], Gee-Ann TOH [7], Raoul Mazars[1], Caio Bomfim[1], Fabienne Hervé [2], Korian Lhaute [2], Damien Réveillon [2], Bastien Suire[1], Léa Ravon-Katossky[1], Thomas Benoist [1], Léa Fromont [1], David Péricat[1], Kenneth Neil Mertens[8], Amélie Derrien [8], Aouregan Terre-Terrillon[8], Nicolas Chomérat[8], Gwenaël Bilien [8], Véronique Séchet [2], Liliane Carpentier[2], Mamadou Fall [9,10], Amidou Sonko[10,11], Hadi Hakim[12], Nfally Sadio[13], Jessie Bourdeaux[14], Céline Cougoule [1], Anthony K Henras [14], Ana Belen Perez-Oliva[6], Patrice Brehmer [11,16,23✉], Francisco J Roca [5,6,17,23✉], Franklin L Zhong [7,18,23✉], John Common [4,15,19,23✉], Etienne Meunier [1,20,23✉] & Philipp Hess [2,21,23✉]

## Abstract

In 2020–2021, a "mysterious illness" struck Senegalese fishermen, causing severe acute dermatitis in over one thousand individuals following exposure through drift-net fishing activity. Here, by performing deep analysis of the environmental samples we reveal the presence of the marine dinoflagellate *Vulcanodinium rugosum* and its associated cyclic imine toxins. Specifically, we show that the toxin PortimineA, strongly enriched in environmental samples, impedes ribosome function in human keratinocytes, which subsequently activates the stress kinases ZAKα and P38 and promotes the nucleation of the human NLRP1 inflammasome, leading to the release of IL-1β/IL-18 pro-inflammatory cytokines and cell death. Furthermore, cell-based models highlight that naturally occurring mutations in the P38-targeted sites of human NLRP1 are unable to respond to PortimineA exposure. Finally, the development and use of human organotypic skins and zebrafish models of PortimineA exposure demonstrate that the ZAKα-NLRP1 axis drives skin necrosis and inflammation. Our results exemplify the threats to human health caused by emerging environmental toxins and identify ZAKα and NRLP1 as important pharmacological targets to mitigate PortimineA toxicity.

**Keywords** Skin Pathology; Environmental Toxins; Ribotoxic Stress Response; NLRP1 Inflammasome
**Subject Categories** Immunology; Microbiology, Virology & Host Pathogen Interaction; Skin

## Introduction

Between November 2020 and 2021, an unexplained skin disease affected artisanal fishermen in Senegal (Kunasekaran et al, 2022; Over 500 fishermen hit by mysterious skin disease in Senegal Reuters). Over a thousand individuals experienced severe acute symptoms, including cutaneous eruptions, fever, and itching, following drift-net fishing activities in an off-shore area South of Dakar (Kunasekaran et al, 2022). The emergence of this perplexing ailment, coupled with its enigmatic origin and unique nature, led national and international media to dub it "mysterious fishermen's illness". Rapid investigations ruled out viral and bacterial infections

[1]Institute of Pharmacology and Structural Biology (IPBS), University of Toulouse, CNRS, Toulouse, France. [2]Ifremer, PHYTOX Physiologie et Toxines des Microalgues Toxiques et Nuisibles, F-44000 Nantes, France. [3]Groupe Rocher, Research-Innovation & Development Department, Issy-les-Moulineaux, France. [4]A*STAR Skin Research, Institute of Singapore, Agency for Science, Technology and Research (A*STAR) Skin Research Labs, 138648 Singapore, Singapore. [5]Department of Biochemistry and Molecular Biology-B and Immunology, Infectious Disease Pathology, Clinical Microbiology and Tropical Medicine, University of Murcia, Murcia, Spain. [6]Biomedical Research, Institute of Murcia (IMIB)-Pascual Parrilla, Murcia, Spain. [7]LKC School of Medicine, Nanyang Technological University, Singapore, Singapore. [8]Ifremer, COAST Unit, Concarneau, France. [9]Université Cheikh Anta Diop de Dakar, Laboratoire de Toxicologie et d'hydrologie, Dakar-Fann, Senegal. [10]Anti-Poison Centre, Fann University Hospital, Dakar, Senegal. [11]Institut de Recherche pour le Développement, IRD, Univ Brest, CNRS, Ifremer, Dakar, Senegal. [12]Private Dermatologist, Dakar, Senegal. [13]Institut Sénégalais de Recherche Agricole, Centre de Recherche Océanographique de Dakar Thiaroye, Dakar, Senegal. [14]Molecular, Cellular and Developmental (MCD) Unit, Centre for Integrative Biology (CBI), CNRS, University of Toulouse, UPS, Toulouse, France. [15]Translational and Clinical Research Institute and NIHR Newcastle Biomedical Research Centre, Newcastle University, Newcastle upon Tyne, UK. [16]Present address: SRFC, Sub regional Fisheries Commission, Liberté 5, Dakar, Senegal. [17]Present address: Department of Biochemistry and Molecular Biology-B and Immunology, Infectious Disease Pathology, Clinical Microbiology and Tropical Medicine, University of Murcia, Murcia, Spain. [18]Present address: A*STAR Skin Research Institute of Singapore, Agency for Science, Technology and Research (A*STAR) Skin Research Labs, 138648 Singapore, Singapore. [19]Present address: A*STAR Skin Research Institute of Singapore, Agency for Science, Technology and Research (A*STAR) Skin Research Labs, 138648 Singapore, Singapore. [20]Present address: Institute of Pharmacology and Structural Biology (IPBS), University of Toulouse, CNRS, Toulouse, France. [21]Present address: Ifremer, PHYTOX Physiologie et Toxines des Microalgues Toxiques et Nuisibles, F-44000 Nantes, France. [22]These authors contributed equally as first authors to this work: Léana Gorse, Loïc Plessis, Stephen Wearne, Margaux Paradis, Miriam Pinilla, Rae Chua. [23]These authors contributed equally as senior authors to this work: Patrice Brehmer, Francisco J Roca, Franklin L Zhong, John Common, Etienne Meunier, Philipp Hess.
✉E-mail: Patrice.Brehmer@ird.fr; fjroca@um.es; franklin.zhong@ntu.edu.sg; john.common@newcastle.ac.uk; etienne.meunier@ipbs.fr; philipp.hess@ifremer.fr

as well as various chemical pollutions, including chlorinated biphenyls, polycyclic aromatic hydrocarbons and pharmaceutical residues (Kunasekaran et al, 2022). The absence of such suspected pollutants from the collected water samples raised our interest in the potential role of microalgae in the incident.

In nature, the massive proliferation of certain microalgae can occur due to favorable environmental conditions, either as a result of natural causes (e.g., nutrient availability or seasonal variations in light and temperature) or human activities (e.g., industrial and agricultural pollution, the introduction of invasive species through maritime transport, or climate change) (Berdalet et al, 2015). Some harmful algae pose a direct threat to human health, either through skin exposure during activities along the coast or at sea or through toxin-containing aerosols on the foreshore (Lim et al, 2023).

Recently, the dinoflagellate *Vulcanodinium rugosum* (*V. rugosum*) was suspected to be responsible for severe skin irritation that occurred in bathing people in Cienfuegos Bay, Cuba (Moreira-González et al, 2021). The bloom caused acute dermatitis in around 60 beachgoers, primarily children. *Vulcanodinium rugosum* was initially taxonomically described in France (WoRMS—World Register of Marine Species—*Vulcanodinium rugosum* Nézan and Chomérat, 2011) but is found worldwide, including in Japan, Australia, New Zealand, China, and the United States of America (Garrett et al, 2014; Zeng et al, 2012; Rhodes et al, 2011). Toxins identified in *V. rugosum* include neurotoxic Pinnatoxins and cytotoxic Portimines (referred to as Portimine A and B), which induce apoptosis in certain cell lines (Selwood et al, 2015; Munday et al, 2012; Smith et al, 2011; Cuddihy et al, 2016; Fribley et al, 2019; Cangini et al, 2024; Bouquet et al, 2023b; Norambuena and Mardones, 2023; Zingone et al, 2021; Abadie et al, 2018; García-Cazorla and Vasconcelos, 2022; Hogeveen et al, 2021; Hort et al, 2023; Sosa et al, 2020). Given the similarity of symptoms reported between the "mysterious fishermen's illness" in Senegal and the incident in Cienfuegos Bay, we hypothesized the presence of *V. rugosum* and its toxins as causative agents. Analysis of environmental samples collected during the outbreak confirmed the presence of the microalgae, thus raising questions about how *V. rugosum* and its toxins contributed to severe cutaneous symptoms in both Cuba and Senegal (Fig. 1A–C).

To cope with harmful stimuli, such as pathogens or other potential threats (e.g., tissue injury, UV radiation, chemicals), the innate immune system has evolved a set of sensors known as pattern recognition receptors (PRRs) (Sundaram et al, 2024; Robinson and Boucher, 2024; Yu et al, 2024; Martinon et al, 2002; Medzhitov, 2021). These receptors, enriched in immune and epithelial cells, play a key role in orchestrating inflammatory responses (Sundaram et al, 2024; Medzhitov, 2021). While essential for an effective immune response, dysregulated inflammatory reactions can lead to severe pathogenic outcomes, ranging from chronic auto-inflammation to irreversible tissue damage (Medzhitov, 2021). Specifically, the inflammasome-forming sensor family constitutes one family of PRRs able to promote both cell death and an inflammatory response upon detection of specific signals, a process that we hypothesized to be occurring during *V. rugosum* exposure (Robinson and Boucher, 2024; Yu et al, 2024; Sundaram et al, 2024). Inflammasomes are intracellular pro-inflammatory complexes that, upon activation by various stress signals or pathogens, promote pyroptotic cell death as well as IL-1β/18 inflammatory cytokines release (Yu et al, 2024; Robinson and

Boucher, 2024). Such process requires the inflammasome-activated protease Caspase-1 which will promote both IL-1β/18 cleavage and maturation as well as Gasdermin D-dependent plasma pore formation and pyroptosis and subsequent Ninjurin-1-driven cell lysis (Yu et al, 2024; Robinson and Boucher, 2024; Newton et al, 2021; Kayagaki et al, 2021; Shi et al, 2015; Kayagaki et al, 2015; Martinon et al, 2002; Sundaram et al, 2024).

Given the highly necrotic response observed in the skin of individuals exposed to *V. rugosum*, we hypothesized that certain essential cellular functions driven by the inflammasomes were disrupted, leading to an acute inflammatory response.

# Results

## *Vulcanodinium rugosum*-produced Portimine A triggers human skin epithelial cell necrosis and IL-1 cytokine release

To investigate the origin of fishermen's dermatitis in Petite Côte, Senegal, samples from various environmental compartments were collected during both the 2020 and 2021 outbreaks, at the coastal sites where the affected fishermen operated (Fig. 1A). Observations revealed a mixture of sediment and unidentified biomass on the fishing canoe bottom and nets (Samples I and II). Light microscopy of Sample I revealed the presence of thecate cells and temporary cysts of the armored dinoflagellate *Vulcanodinium rugosum* (Fig. 1B) while quantitative PCR analysis confirmed the presence of *V. rugosum* in Sample I (Fig. EV1A). Further analysis of Sample I using Liquid Chromatography coupled to tandem Mass Spectrometry (LC-MS/MS), showed the presence of five specific *V. rugosum* cyclic imine toxins, namely Portimines (A and B) and Pinnatoxins (-H, -H Iso, and -G) (Fig. 1C). The quantification of these toxins in the environmental samples indicated that Portimine A was largely predominant in each sample (Fig. 1C).

Next, to determine if seawater containing cyclic imine toxins was responsible for the inflammatory skin response observed in fishermen, we evaluated their inflammatory potential on primary human skin keratinocytes (pHEKs) by measuring inflammatory cytokine release. Exposure of pHEKs to extracts derived from samples obtained from net biomass (here referred to as Sample II) led to the detection of a strong inflammatory response characterized by an enrichment of alarmins and cytokines of the IL-1 family, such as IL-1α, IL-1β, and IL-18 (Figs. 1D and EV1B). In addition, Sample II induced cell lysis in treated keratinocytes, likely resulting from necrosis as assessed by LDH release (Fig. 1E). This cell death was inhibited by the pan-caspase inhibitor Z-VAD and, partly inhibited by caspase-1 inhibitors Z-YVAD and VX-765, suggesting that one or multiple cyclic imine toxins present in extracts from sample II triggered both keratinocyte pyroptosis and IL-1 family cytokine release (Fig. 1E).

Next, to determine the respective contribution of each toxin to these phenotypes, we assessed keratinocyte death and release of IL-1 family cytokines after treatment with purified Pinnatoxins G/H, Portimine A or its analog Portimine B. We observed that pHEKs underwent plasma membrane permeabilization (SYTOX Green incorporation), cell lysis (LDH release) as well as released IL-1β/IL-18 upon exposure specifically to Portimine A/B, but not to Pinnatoxins G/H (Figs. 1F and EV1C–F), indicating that Portimine

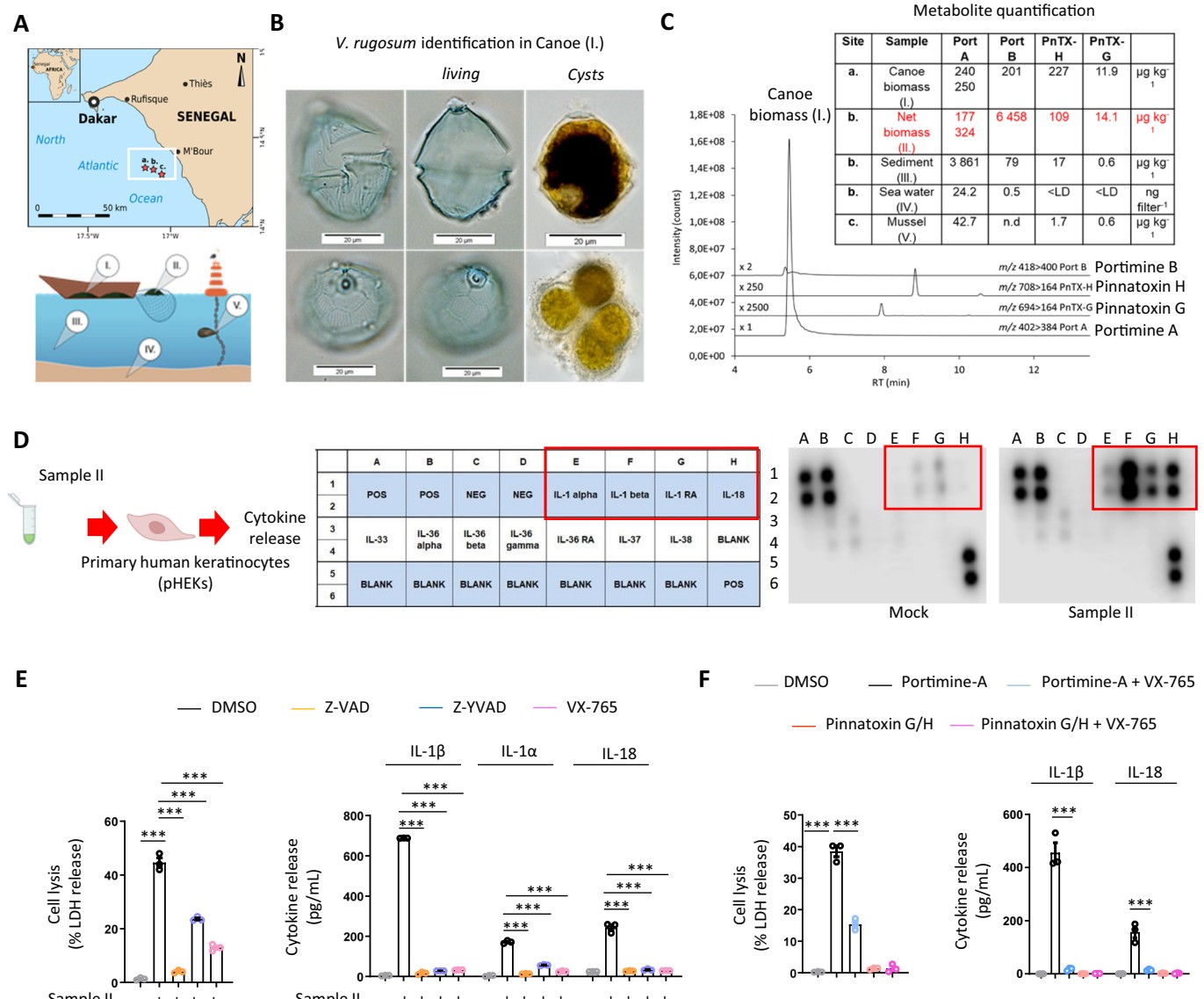

**Figure 1.** *Vulcanodinium rugosum*-produced Portime A triggers human skin epithelial cell necrosis and IL-1 cytokine release.

(A) Map of the Senegalese coast showing the location of the outbreak and the sampling sites i.e., red star (a) (14.33 N; −17.15 W), (b) (14.32 N; −17.10 W) and (c) (14.29 N; −17.05 W). Samples from different environmental compartments: (I) biomass from fishing canoe, (II) biomass from fishing drift-net, (III) GF/F filtered seawater, (IV) marine sediment, (V) mussel flesh. (B) Light microscopy images of *Vulcanodinium rugosum* (sample I, 2020): living cells and temporary cysts. (C) LC-MS/MS chromatogram of toxin profile (sample I, 2020) and chemical structures of Portime A and Pinnatoxin H and concentrations of *V. rugosum* toxins in the samples I–V, quantified by LC-MS/MS. N.B. All mussel (c) data are from 2021. PnTX: Pinnatoxin, Port: Portimine. (D) Cytokine analysis 24 h after exposure of primary human keratinocytes (pHEKs) to purified extracts derived from Sample II isolated in (C, red), diluted 1/20,000. Representative experiment of three independent replicates. (E) Quantification of cell lysis (LDH) and IL-1α, IL-1β, or IL-18 release in pHEKs treated with Sample II (1/20,000) for 24 h. When specified, the pan-caspase inhibitor (Z-VAD, 20 μM), Caspase-1 inhibitors (Z-YVAD, 20 μM or VX-765, 10 μM) were used. ***$P \le 0.0001$, two-way ANOVA with multiple comparisons. Values are expressed as mean ± SEM. Graphs show one experiment performed in triplicates at least three times. (F) Cell lysis (LDH) and IL-1β or IL-18 release evaluation in pHEKs upon pure Portime A (4 ng/mL) or Pinnatoxins-H/G (40 ng/mL) exposure for 24 h. When specified, the -caspase inhibitor (Z-VAD, 20 μM), Caspase-1 inhibitors (Z-YVAD, 20 μM or VX-765, 10 μM) were used. ***$P \le 0.0001$, two-way ANOVA with multiple comparisons. Values are expressed as mean ± SEM. Graphs show one experiment performed in triplicates at least three times. Source data are available online for this figure.

A or B could be the toxin responsible for the skin inflammation observed in contaminated areas. In addition, the use of the Caspase-1 inhibitor VX-765 completely abrogated IL-1β/IL-18 release and reduced PortimineA-induced cell lysis of pHEKs as we observed with extracts from Sample II (Fig. 1F). Furthermore, pure toxins PnTX-G and -H, Portimine A and B, as well as extracts from

cultured *V. rugosum* and Senegalese biomass (samples I and II) were tested for pHEK cytotoxicity (MTT assay). Portimine A exhibited a ca. 20-times lower $IC_{50}$ compared to Portimine B, i.e., significantly greater toxicity with an $IC_{50}$ of around 1 nM (Fig. EV1C,D). By expressing the extract concentration in PortimineA equivalents (determined by LC-MS/MS), their dose-

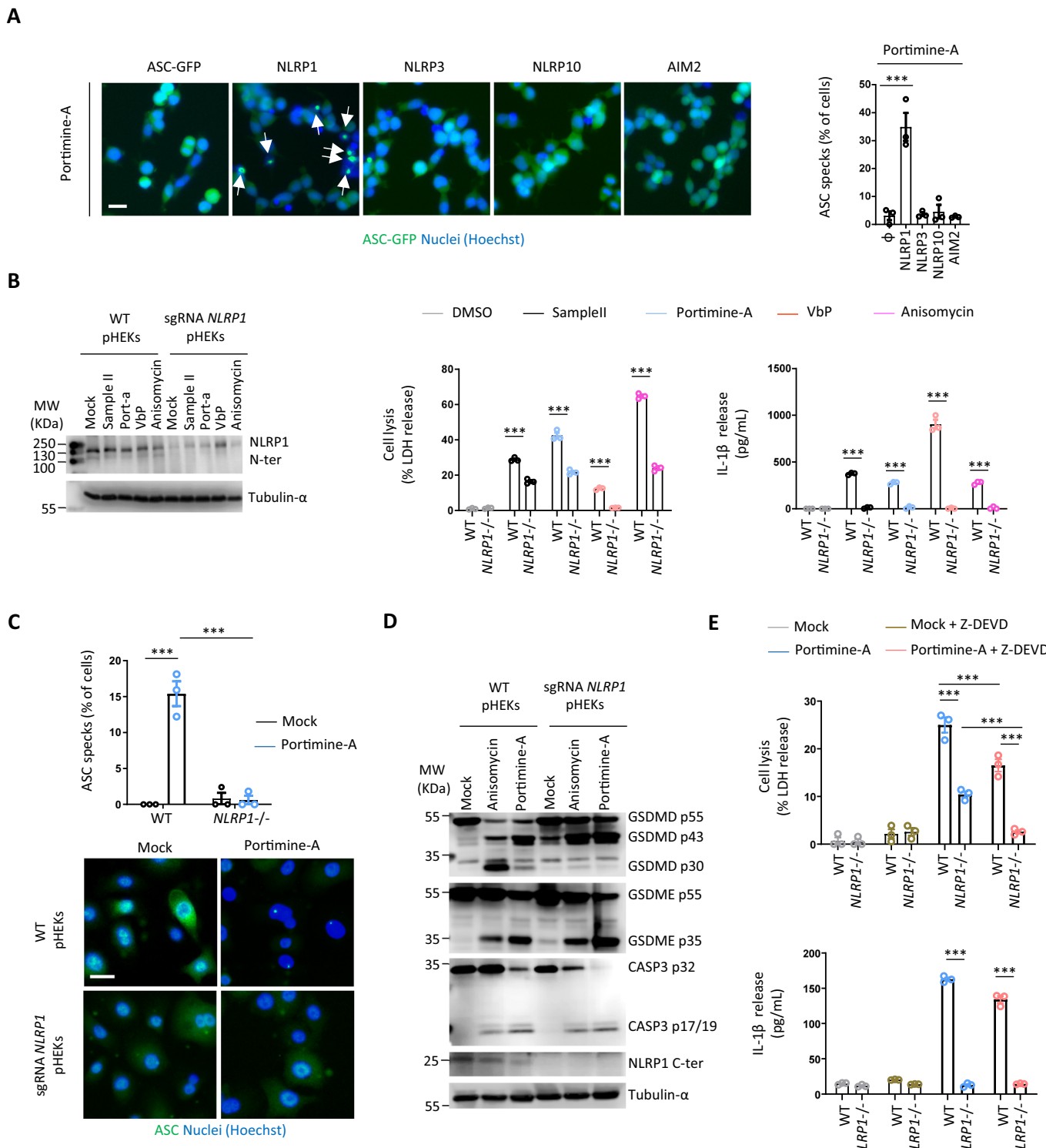

response curves were very similar to that of Portimine A, obtaining IC$_{50}$-values in a comparable range, i.e., between 1.07 and 1.71 nM (Fig. EV1D). These results strongly suggested that the cytotoxic effects of the extracts may be primarily driven by the presence of Portimine A, potentially masking the weaker activity of Portimine B, which is present at significantly lower concentrations (N.B.

Portimine B is found 27- to 50-fold less concentrated than Portimine A in environmental samples (Fig. 1C)). Relying on the IC$_{50}$ observed with Sample II (between 1.07 and 1.71 nM) (Fig. EV1D), we generated mixtures of Pinnatoxins G/H containing 1.7 nM of Portimine A or of 1.7 nM of Portimine B or with Pinnatoxins G/H + 1.7 nM Portimine A + 1.7 nM Portimine B and

**Figure 2. Portimine A activates the NLRP1 inflammasome in human skin epithelial cells.**

(**A**) Fluorescence micrographs and associated quantifications of ASC-GFP specks in HEK293T cells individually expressing or not NLRP1, NLRP3, NLRP10 or AIM2 exposed to 4 ng/mL of Portimine A for 8 h. ASC-GFP (green) pictures were directly taken after adding Hoechst (nuclei staining). Images shown are from one experiment and are representative of $n = 3$ independent experiments. Scale bar, 10 μm. The percentage of ASC complex was performed by determining the ratios between cells positive for ASC speckles and the total of cell nuclei (Hoechst). At least ten fields from each experiment were analyzed. Values are expressed as mean ± SEM. ***$P \leq 0.0001$, one-way ANOVA. (**B**) Immunoblotting characterization of the NLRP1 genetic knockdown (CRISPR-Cas9) and of the subsequent cell lysis (LDH) and IL-1β release in pHEKs exposed or not to Sample II (1/20,000 dilution), Portimine A (4 ng/mL), ValboroPro (VbP, 10 μM) or Anisomycin (1 μM) for 24 h. For Cell lysis and cytokine release, ***$P \leq 0.0001$, two-way ANOVA with multiple comparisons. Values are expressed as mean ± SEM. Immunoblot is one experiment representative of three independent experiments. Graphs show one experiment performed in triplicates at least three times. (**C**) Fluorescence micrographs and associated quantifications of ASC specks in pHEKs and NLRP1-deficient pHEKs generated in (**B**) and exposed or not to Portimine A (4 ng/mL) for 24 h. Hoechst (nuclei staining), ASC (anti-ASC antibody, green). Images shown are from one experiment and are representative of $n = 3$ independent experiments. Scale bar, 10 μm. The percentage of ASC complex was performed by determining the ratios between cells positive for ASC speckles and the total of cell nuclei (Hoechst). At least ten fields from each experiment were analyzed. Values are expressed as mean ± SEM. ***$P \leq 0.0001$, one-way ANOVA. (**D**) Immunoblotting of GSDMD, GSDME, Caspase-3, NLRP1 (C-term part) and tubullin in pHEKs and NLRP1-deficient pHEKs (generated in (**B**)) after 24 h exposure to Portimine A (4 ng/mL) or to the known RSR inducer Anisomycin (1 μg/mL). Immunoblots show lysates from one experiment performed at least two times. (**E**) Cell lysis (LDH) and IL-1β release evaluation in pHEKs and NLRP1-deficient pHEKs upon pure Portimine A (4 ng/mL) or Anisomycin (1 μg/mL) exposure for 24 h. When specified, the Caspase-3 inhibitor (Z-DEVD, 20 μM) was used. ***$P \leq 0.0001$, two-way ANOVA with multiple comparisons. Values are expressed as mean ± SEM. Graphs show one experiment performed in triplicates at least three times. Source data are available online for this figure.

analyzed cell lysis (LDH release) in pHEKs (Fig. EV1E). The results showed that only the mixtures with Portimine A induced cell lysis in conditions mimicking the concentration of each toxin analyzed in the Sample II, which suggests that Portimine B exhibits very low activity compared to Portimine A for yet-to-be determined reason (Fig. EV1E). This is in agreement with recent findings from the Baran group that also determined a very low toxic potency of chemically synthesized Portimine B on various cell lines (Tang et al, 2023) and our own observations that Portimine B needs to be strongly concentrated to induce detectable keratinocyte death (Fig. EV1C–F). Thus, both the low concentration of Portimine B in the environmental samples and its extremely low potency at triggering cell death pointed to Portimine A as a putative causative agent responsible of skin inflammation and damage observed in fishermen.

Finally, we determined whether PortimineA exposure could also target other cell types. We evaluated cell death of primary human endothelial and nasal epithelial cells, and blood-isolated monocytes, neutrophils and lymphocytes. Portimine A induced cell death only in endothelial and nasal epithelial cells, suggesting that blood-isolated monocytes, neutrophils and lymphocytes exhibit some intrinsic protection against this toxin (Fig. EV1G). Again, Portimine B failed at triggering cell death at similar concentration range than Portimine A (Fig. EV1G), thus arguing in favor of a major contribution of Portimine A over Portimine B at promoting cell death.

Thus, our results identify *V. rugosum*-derived Portimine A as the probable causative agent of the inflammatory skin lesions observed in Senegalese fishermen.

## Portimine A activates the NLRP1 inflammasome in human skin epithelial cells

Caspase-1-induced IL-1β and IL-18 release and cell pyroptosis require inflammasome activation. Thus, we tested whether one or several inflammasomes could be engaged in pHEKs upon PortimineA exposure. To achieve this, we generated reporter human HEK293 cell lines by expressing the ASC-GFP adaptor construct, and we assessed inflammasome complex formation, called ASC specks (bright quantifiable puncta, (Pinilla et al, 2023)) by fluorescence microscopy in the presence of individual inflammasome sensors previously reported to be expressed in pHEKs (NLRP1, NLRP3, AIM2 and NLRP10) (Robinson and Boucher, 2024). Only NLRP1-expressing cells responded to Portimine A by assembling inflammasome complexes (determined by ASC speck formation) (Fig. 2A). Furthermore, pure toxins PnTX-G and -H failed at triggering ASC specks in hNLRP1-expressing cells and Portimine B starting triggering few ASC speck formation at extremely high concentrations (400 ng/mL, 100-fold PortimineA concentration used) (Fig. EV2). These results suggest that NLRP1 could be the sensor triggering the cell responses observed in PortimineA-treated pHEKs. To corroborate our findings, we genetically deleted *NLRP1* from pHEKs using CRISPR/Cas9 technology and exposed those cells to Portimine A, Sample II extract or the well-known NLRP1 activators Val-boro-Pro (VbP) and Anisomycin (Fig. 2B). Portimine A and Sample II failed to induce release of IL-1β/IL-18 cytokines in *NLRP1*-deficient cells (Fig. 2B). In addition, NLRP1 deficiency partly protected cells against Sample II- and PortimineA-induced cell death, suggesting that the human NLRP1 inflammasome drives both cell IL-1β/IL-18 release and pyroptosis in keratinocytes in response to Portimine A (Fig. 2B). Further analysis of ASC speck formation in pHEKs invalidated or not for *NLRP1* showed that only wild-type (WT) pHEKs assembled ASC specks in response to Portimine A, confirming the relevance of NLRP1 as a sensor responding to PortimineA exposure (Fig. 2C).

Finally, we aimed at determining the identity of the NLRP1-independent cell death pathway engaged in pHEKs upon PortimineA exposure. As Z-VAD, a pan-caspase inhibitor, abrogated PortimineA-induced keratinocyte death in response to Portimine A, we speculated that NLRP1- and Caspase-1 independent cell death might be triggered by another Caspase. By performing Immunoblotting in NLRP1 and NLRP1-deficient keratinocytes, we observed that the protease Caspase-3 and its substrate Gasdermin E were strongly activated in a NLRP1-independent manner in response to Portimine A (Fig. 2D). To the contrary, the Caspase-1-generated p30 activate fragment of Gasdermin D was found to depend on NLRP1 upon exposure to Portimine A or Anisomycin, a well know activator of the NLRP1 inflammasome (Fig. 2D). This suggested to us that Caspase-3 could be involved in the remaining cell death observed in the NLRP1-deficient keratinocytes. In this context, treatment of

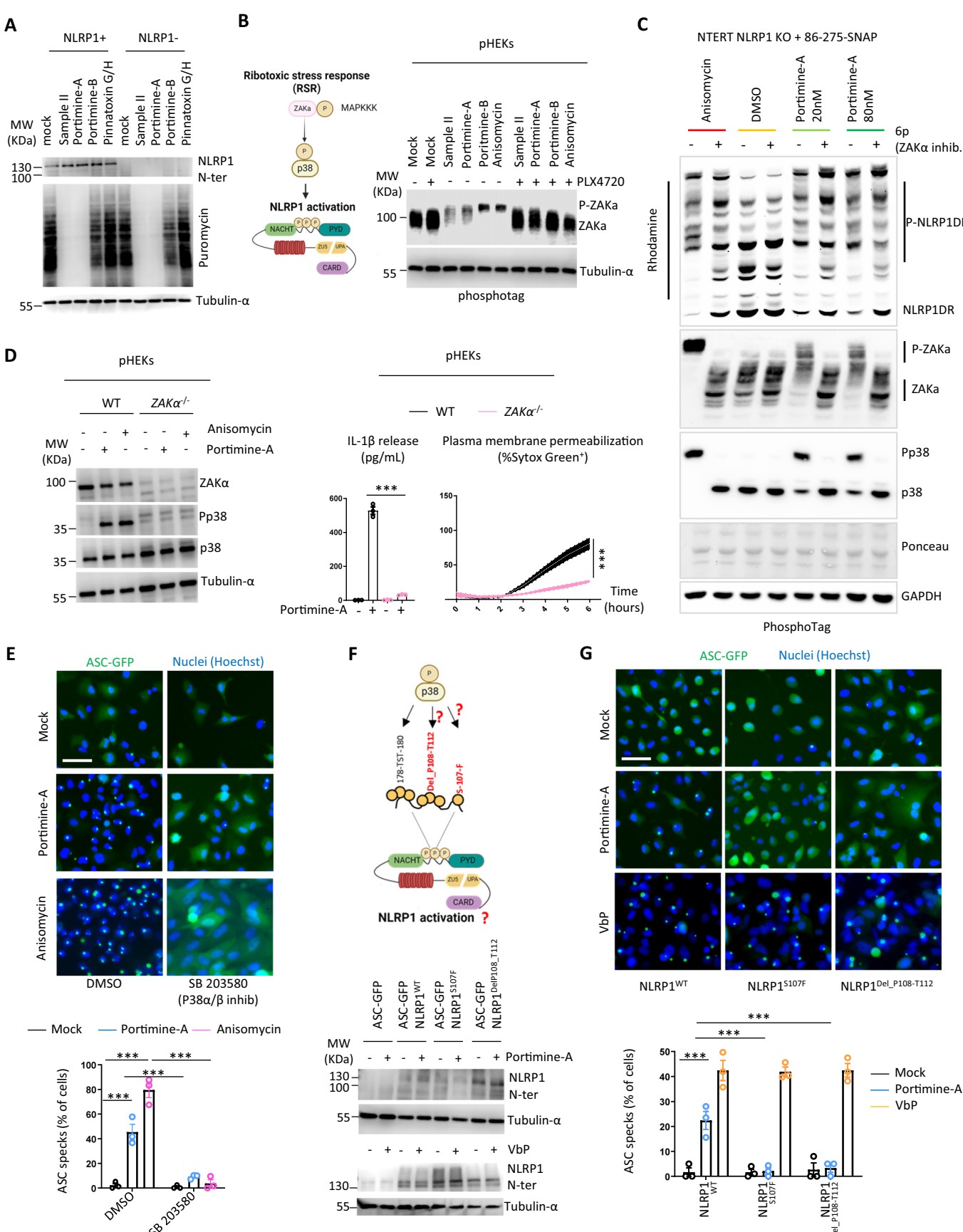

**Figure 3. Portimine-inhibited translation promotes ZAKα-dependent P38 activation and hNLRP1 inflammasome activation in epithelial cells.**

(A) Determination of protein synthesis in HEK293 cells expressing or not NLRP1 in response to Sample II (1/20,000 dilution), Portimine A (4 ng/mL), Portimine B (400 ng/mL), or Pinnatoxins-H/G (40 ng/mL) by measuring puromycin incorporation after 2 h exposure. Immunoblots show lysates from one experiment performed at least three times. (B) Schematic representation of the mechanism of ZAKα/P38 stress kinases activation upon induction of Ribotoxic Stress Response (RSR). Phosphotag blotting of phosphorylated ZAKα in pHEK cells exposed to Sample II (1/20,000 dilution), Portimine A (4 ng/mL), Portimine B (400 ng/mL), or the known RSR inducer Anisomycin (1 μg/mL) for 8 h. When specified, PLX420 (b-Raf, ZAKα inhibitor, 10 μM) was used. Immunoblots show lysates from one experiment performed at least three times. (C) Phosphotag blotting of phosphorylated ZAKα, P38 and NLRP1 disordered Region (DR) in NTERT NLRP1 KO + 86-275-SNAP (described in Fig. EV3D) cells exposed to Portimine A (20 or 80 ng/mL) or to the known RSR inducer Anisomycin (1 μg/mL) for an hour. When specified, 6p (ZAKα inhibitor, 1 μM) was used. Ponceau staining and GAPDH were used as internal protein loading controls. Immunoblots show lysates from one experiment performed at least two times. (D) Immunoblotting of P38, ZAKα, Tubulin, and phosphorylated P38, plasma membrane permeabilization (SYTOX Green incorporation, 6 h) and IL-1β release evaluation (24 h) in pHEKs WT or genetically invalidated (CRISPR-Cas9) for ZAKα 8 h after exposure to Portimine A (4 ng/mL) or the known RSR inducer Anisomycin (1 μg/mL). ***$P \leq 0.0001$, one-way ANOVA. Values are expressed as mean ± SEM. Graphs show one experiment performed in triplicates at least three times. (E) Fluorescence microscopy and associated quantifications of ASC-GFP specks in A549$^{ASC-GFP}$ reporter cells expressing or not NLRP1 exposed to 4 ng/mL of Portimine A or to 1 μg/mL of Anisomycin for 6 h. When specified, SB 203580 (P38α/β inhibitor, 10 μM) was used. ASC-GFP (green) pictures were directly taken in dish after adding Hoechst (nuclei staining). Images shown are from one experiment and are representative of three independent experiments; scale bars, 10 μm. ASC complex percentage was performed by determining the ratios of cells positive for ASC speckles on the total nuclei (Hoechst). At least ten fields from each experiment were analyzed. Values are expressed as mean ± SEM. ***$P \leq 0.0001$, one-way ANOVA. (F) Western blot showing NLRP1 using an anti-NLRP1 N-terminal antibody (aa 1–323) in HEK293$^{ASC-GFP}$ reporter cells reconstituted with hNLRP1 or hNLRP1 plasmid constructs mutated (S107F and DelP108_T112) after 6 h exposure to Portimine A (4 ng/mL) or after 10 h exposure to Val-boro-Pro (VbP, 10 μM). Images shown are from one experiment and are representative of three independent experiments. (G) Fluorescence micrographs and respective quantifications of ASC-GFP specks in HEK293$^{ASC-GFP}$ reporter cells reconstituted with hNLRP1 or hNLRP1 plasmid constructs mutated for S107F and DelP108_T112 after 6 h exposure to Portimine A (4 ng/mL) or after 10 h exposure to Val-boro-Pro (VbP, 10 μM). ASC-GFP (green) pictures were taken in the dish after toxin exposure. Images shown are from one experiment and are representative of three independent experiments; scale bars, 10 μm. ASC complex percentage was performed by determining the ratios of cells positive for ASC speckles (green, GFP) on the total nuclei (Hoechst). At least ten fields from three independent experiments were analyzed. Values are expressed as mean ± SEM. ***$P \leq 0.0001$, two-way ANOVA with multiple comparisons. Graphs show one experiment performed in triplicate at least three times. Source data are available online for this figure.

NLRP1-deficient keratinocytes with Z-DEVD, a Caspase-3 inhibitor, entirely abrogated PortimineA-induced cell death (Fig. 2E), suggesting that a Caspase-3-dependent, yet NLRP1-independent, cell death program is also induced by Portimine A in keratinocytes.

All in one, our results point to the human NLRP1 inflammasome in keratinocytes as a major player in the inflammatory response triggered by Portimine A but also suggest that Portimine A can drive additional cell death programs.

## Portimine-inhibited translation promotes ZAKα-dependent P38 activation and hNLRP1 inflammasome activation in epithelial cells

As we identified *V. rugosum*-produced Portimine A as the trigger for NLRP1-mediated inflammasome responses in pHEKs, we next aimed to decipher the molecular events driving NLRP1 activation. We first interrogated ribotoxic stress response (RSR) as this mechanism of ribosome inactivation has emerged as a critical link leading to NLRP1 inflammasome-mediated responses in humans (Robinson et al, 2022). In this context, we performed Puromycin incorporation assays as a readout of protein translation in cells (Pinilla et al, 2023). We found that Sample II, Portimine A and to a very lower extent its analog Portimine B, but not Pinnatoxin G/H, inhibited protein translation in HEK293 reporter cell lines or in pHEKs (Figs. 3A and EV3A). Importantly, this inhibition was not reversed by the lack of NLRP1 expression, showing that NLRP1 inflammasome activation is downstream of PortimineA-inhibited protein translation. We next determined if Portimine A could directly impair ribosome function. First, we found an accumulation of free 60S subunits and 80S ribosomes accompanied by a decrease in polysome formation in PortimineA-treated cells, indicating that Portimine A impairs ribosome function (Fig. EV3B). The presence of ribosomal protein RPS6, a polysome marker, was decreased in the different fractions containing ribosomes and polysomes,

corroborating a decrease in ribosome activity in PortimineA-treated cells (Fig. EV3B). Recent work from (Tang et al, 2023) showed that Portimine A could interact with NMD3, a regulator of ribosomal 60S subunit synthesis. Addressing the presence of the regulators NMD3 and EIF6 in fractions containing the free 60S subunits and 80S ribosomes, we found no change of NMD3/EIF6 in PortimineA-treated fractions (Fig. EV3B), hence suggesting that inhibition of ribosome function by Portimine A was not due to altered NMD3 recruitment to the 60S subunit. Next, to determine if Portimine A directly inhibits ribosome activity, we used a model of in vitro translation by rabbit reticulocyte lysates (Olsnes et al, 1973). Exposure of reticulocyte lysates to Sample II, Portimine A, or Anisomycin (positive control to cause inhibition of translation) all impaired translation of the reporter protein Interleukin (IL)-33, hence suggesting that PortimineA-inactivated translation was caused by directly targeting ribosome function (Fig. EV3C).

Then, we asked about the downstream mechanisms by which PortimineA-inactivated ribosome function could lead to NLRP1 inflammasome-mediated responses. It has been shown that ribosome stalling and/or collision promotes the activation of the apical stress kinase of the MAPK family, ZAKα, a process essential for the activation of P38 stress kinases and subsequent NLRP1 phosphorylation in a specific linker (referred here to as Disordered Region, DR, aa 86 to 275) and activation (Vind et al, 2020; Wu et al, 2020; Robinson et al, 2022) (Fig. 3B). Thus, we first tested the ability of Sample II, Portimine and high doses of Portimine B (400 ng/mL) to trigger ZAKα phosphorylation and in turn activation, using Anisomycin as a positive control of ZAKα activation. We found that all these treatments induced a robust ZAKα phosphorylation, which was inhibited in the presence of PLX4720 which inhibits ZAKα activity (Fig. 3B). ZAKα activates P38 kinases, which target multiple Serine (S) and Threonine (T) residues in the hNLRP1 linker called Disordered Region (DR, aa 86-275), including S 107, TST 112–114, and TST 178–180, all

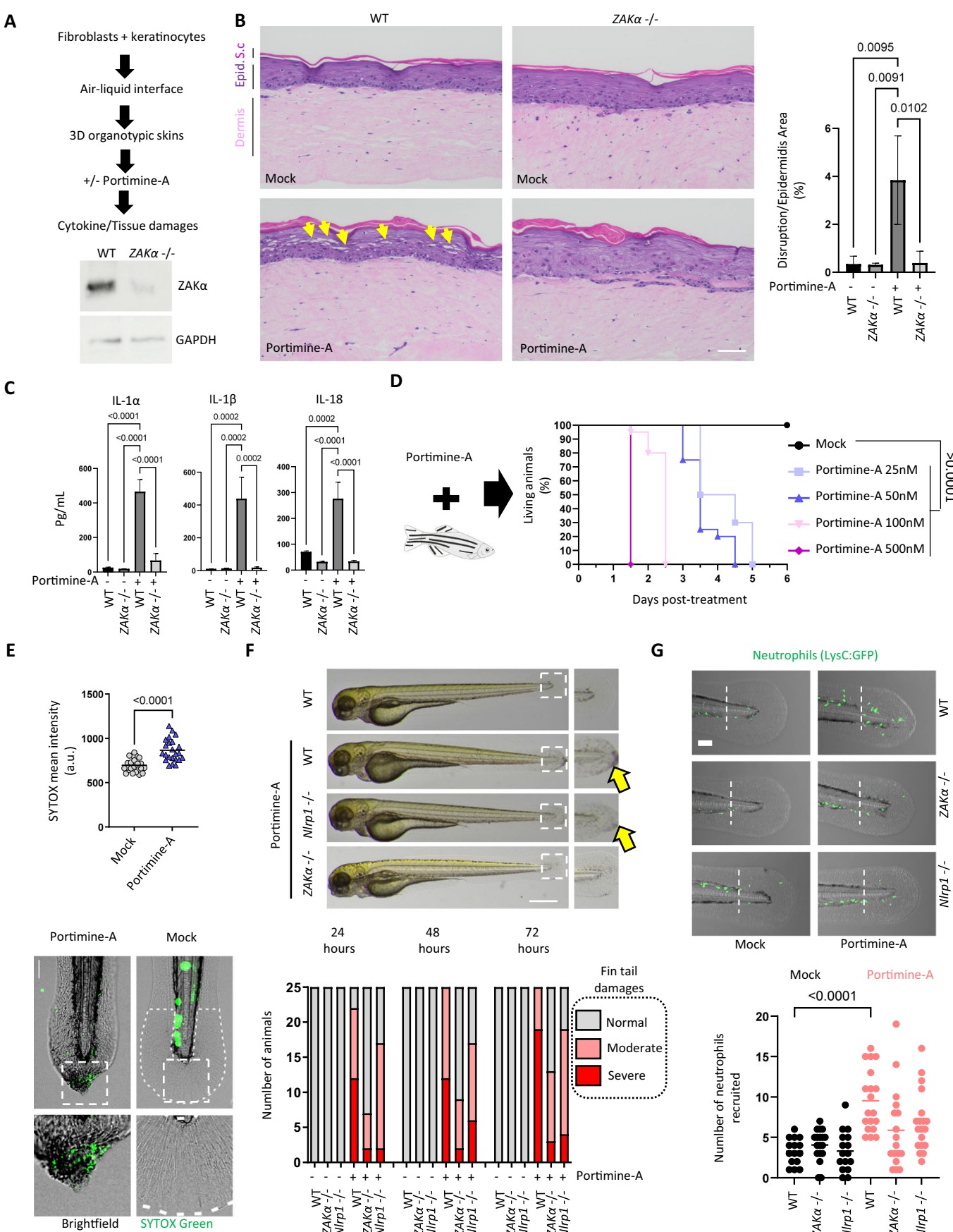

**Figure 4. ZAKα and NLRP1 contribution to Portimine-induced skin inflammation in 3D skin and zebrafish models.**

(A) Representative diagram indicating the experimental approach in the WT and ZAKα-deficient 3D skin model of Portimine exposure. (B) Hemalun (H) & Eosin (E) staining showing ZAKα-dependent histological changes caused by PortimineA (20 nM) exposure. Yellow arrows show epidermidis alterations/damages induced by Portimine A. Associated quantification of the dermal–epidermal layer detachment of 3D skin. P values indicated in figure, one-way ANOVA. Images are representative of three biological replicates. Scale bar = 50 µm. (C) IL-1α, β, and -18 cytokine analysis in 3D skin treated or not with Portimine A (A, B). Results are from a total of 65 cytokine analysis presented in Fig. EV4A. P values indicated in figure, one-way ANOVA. (D) Survival curves of zebrafish larvae (20 larvae/group) incubated with various concentrations of Portimine A. Graph shows a representative experiment out of the three performed. P values indicated in figure, Log-Rank (Mantel–Cox) test. (E) Determination of skin necrosis induced by Portimine A (25 nM) in the caudal fin of zebrafish larvae by measuring the incorporation of the plasma membrane impairment probe SYTOX Green after 24 h exposure in area delimited by dashed white line. In total, 20–30 larvae/group were compared and quantified. Values are expressed as mean. P values indicated in figure, one-way ANOVA. Scale bar 100 µm. Graphs show one experiment performed three times. (F) Determination and quantification of zebrafish larvae fin tail gross damage induced by Portimine A (25 nM) in WT and Nlrp1- and ZAKα-deficient larvae (20/group) at the indicated time points. Scale bar 100 µm. Specific quantifications and statistical analysis are provided in experiments that measure tail area presented Fig. EV4B,D. Graphs show one experiment performed two times. (G) Determination and quantification of zebrafish neutrophil recruitment to the fin tail in WT and Nlrp1- and ZAKα-deficient larvae (20/group) of the zebrafish line Tg(LysC:GFP)^nz117 with GFP-expressing neutrophils over 24 h exposure to Portimine A (75 ng/mL). Graph shows a representative experiment out of the two performed. Values are expressed as mean. P values indicated in figure, one-way ANOVA (Kruskal–Wallis test). Scale bar 100 µm. Graphs show one experiment performed two times. Source data are available online for this figure.

essentials for efficient hNLRP1 inflammasome assembly in response to RSR (Robinson et al, 2022; Jenster et al, 2023; Pinilla et al, 2023). Here, we asked if ZAKα activation could lead to NLRP1 DR phosphorylation leading to NLRP1 inflammasome nucleation upon PortimineA exposure by assessing the phosphorylation status of an NLRP1 DR coupled to SNAP tag construct in NLRP1-deficient N/TERT keratinocytes cells in the presence of Portimine A and Anisomycin. We found that both Portimine A and Anisomycin induced efficient NLRP1 DR phosphorylation in addition to activating ZAKα and P38 kinases (Figs. 3C and EV3D). In addition, the use of another ZAKα inhibitor (6p) strongly inhibited ZAKα-dependent P38 activation and NLRP1 DR phosphorylation in response to Portimine A and Ansomycin (Figs. 3C and EV3D). Supporting this, genetic depletion of ZAKα in pHEKs completely impaired Portimine A-induced P38 phosphorylation, IL-1β release and cell death (Fig. 3D). Finally, the P38α/β specific inhibitor SB 203580 also inhibited the assembly of the NLRP1 inflammasome complex and in turn ASC speck formation in reporter cells exposed to Portimine A (and to Anisomycin) (Fig. 3E), hence confirming the critical involvement of the ZAKα-P38 pathway in PortimineA-induced NLRP1 inflammasome activation.

We next tested if naturally occurring NLRP1 variants could influence the response to PortimineA exposure. The gnomAD describes novel mutations in patients on NLRP1 Serine (S)107 (S107F) and in-frame deletion of (DelP108_T112) (NLRP1 | gnomAD v4.1.0 | gnomAD), two important sites phosphorylated by ZAKα and P38 kinases and which trigger hNLRP1 inflammasome activation (Fig. 3F). Expression of WT NLRP1, S107F NLRP1, and DelP108_T112 NLRP1 in ASC-GFP reporter cells showed that those variants were unable to assemble an inflammasome complex and induce ASC speck formation in response to Portimine A while they all assembled ASC specks in response to VbP that activates the NLRP1 inflammasome independently of ZAKα and P38 kinases (Fig. 3G). These results indicate that those naturally occurring mutations in NLRP1 might strongly alter its response to RSR in keratinocytes.

Finally, in order to determine the conservation of ZAK and P38 pathways in two other Portimine A -sensitive cell types, namely basal airway and endothelial cells, we determined their cell lysis in the presence or absence of ZAKα inhibitors upon Portimine A and Anisomycin exposure (Fig. EV3E). We observed that ZAKα

inhibition strongly impaired cell death in response to both Portimine A and Anisomycin, suggesting that ZAKα -driven cell death is conserved in those cell types (Fig. EV3E). To the contrary, exposure of monocytes that show resistance to Portimine A showed no P38 phosphorylation upon PortimineA exposure while Anisomycin triggered efficient ZAKα- and P38-dependent monocyte death (Fig. EV3F), suggesting that the strong difference of response to Portimine A between immune cells and epithelial/endothelial cells might arise upstream from ZAKα-P38 pathway.

Altogether, these results show that PortimineA-induced keratinocyte death and IL-1 cytokine release is driven by ribosome inactivation and subsequent ZAKα-induced P38-dependent hNLRP1 inflammasome activation. In addition, our results also suggest that Portimine A might fail at activating the hNLRP1 inflammasome in people carrying hNLRP1 variants S107F and DelP108_T112.

## ZAKα inactivation impedes Portimine-induced skin inflammation in 3D skin and zebrafish models

Finally, we determined the relevance of the ZAKα-NLRP1 axis on the initiation and development of the inflammatory response to Portimine A in more complex settings: human organotypic skin and a zebrafish model (Rodríguez-Ruiz et al, 2023). We generated wild-type (WT) and ZAKα-deficient human skin organoids by CRISPR/Cas9 technology (Fig. 4A) and found that Portimine A induced significant intraepidermal adhesion loss and cell death (in WT but not in ZAKα-deficient organotypic skins (Fig. 4B)). In addition, from a panel of 65 cytokines tested, IL-1α, IL-1β and IL-18 were the most enriched in the culture medium from PortimineA-stimulated WT organotypic skins (Figs. 4C and EV4A). In contrast, the levels of these cytokines in ZAKα-deficient organotypic skins treated with Portimine A were not different from untreated controls (Fig. 4C). These results confirm the importance of ZAKα at promoting inflammasome-dependent skin inflammation in response to Portimine A.

Subsequently, we took advantage of the partial conservation of the ZAKα-NLRP1 pathway between zebrafish and humans but not with rodents (Li et al, 2018; Rodríguez-Ruiz et al, 2023) to test PortimineA-induced skin inflammation in an animal model. PortimineA administration by soaking drastically decreased survival of zebrafish larvae at 25 nM and higher concentrations

(see material and methods for details) (Fig. 4D). We did not see any detrimental effect in animals treated with 10 nM Portimine A. Interestingly, mortality was associated and preceded by skin damage assessed as gross damage (Fig. EV4B) and presence of cells undergoing necrosis in the caudal fin (Fig. 4E).

Finally, we took a genetic approach to abrogate ZAKa (the zebrafish counterpart of human ZAKα), NLRP1 or the inflammasome adaptor ASC expression in zebrafish larvae through CRISPR/Cas9 technology to evaluate their respective importance on the inflammatory pathology triggered by Portimine A (Fig. EV4C) (Li et al, 2018; Rodríguez-Ruiz et al, 2023). We found that ZAKa deficiency strongly reduced necrosis in the skin of zebrafish caudal fin (Fig. EV4D). We also noticed that NLRP1- and Asc-deficient strains differed from ZAKa-deficient strains as they had less marked, yet significant, impact on skin damages (e.g., s fin tail area and damages) induced by Portimine A (Figs. 4F and EV4D). This suggests and confirms that, as observed in previous work and in this study, ZAKa has a broader function in PortimineA-induced skin damages by promoting both NLRP1-dependent inflammasome response and non-NLRP1-dependent cell death (such as Caspase-3-induced cell death) and tissue damages (Robinson et al, 2023). Finally, in order to determine the importance of ZAKa and NLRP1 on the inflammatory immune cell recruitment in response to PortimineA exposure, we genetically invalidated ZAKa and NLRP1 in the LysC:GFP transgenic line and determined the neutrophil recruitment in the caudal fin (Fig. 4G). We observed that both ZAKa and NLRP1 deficiency reduced neutrophil recruitment in Portimine A-treated fishes (Fig. 4G), which suggests that ZAKa and NLRP1 both play an important role in driving neutrophil recruitment in response to Portimine A.

Altogether, our results suggest a critical role of ZAKα at driving pathological inflammation upon PortimineA exposure both in human 3D skin models and in zebrafish. Furthermore, while ZAKα plays a dominant and apical function in response to PortimineA-induced RSR, NLRP1 partly contributes to this process, suggesting that additional pathways induced by ZAKα also cooperatively contribute to RSR-induced pathology.

# Discussion

Here, we evaluated the causes of the increased skin pathology induced by the microalga *V. rugosum* (Hogeveen et al, 2021; Hort et al, 2023; Tang et al, 2023). The importance of specifically studying this dinoflagellate and its toxins relies on two observations. First, marine ecosystems are warming, acidifying and deoxygenating as a consequence of global warming and these changes are accompanied by increasing impacts of harmful algal blooms on these ecosystems (Griffith and Gobler, 2020), and second, very recently, *V. rugosum* blooms have— for the first time—been associated with skin inflammation and necrosis in exposed bathing people and fishermen ((Moreira-González et al, 2021) and this study)). In this regard, our study unveils that Portimine A promotes human NLRP1 inflammasome activation in skin epithelial cells, leading to pyroptosis and IL-1 family cytokine maturation and release. Mechanistically, activation of the human NLRP1 requires PortimineA-induced ribotoxic stress response (RSR) and subsequent ZAKα-mediated activation of P38 kinases. Although ZAKα-P38-induced NLRP1 inflammasome activation triggers a robust induction of pyroptosis, our results, in agreement with previous studies, also indicate a ZAKα-dependent, yet NLRP1-independent pathway, by which Portimine A triggers cell death through activation of

Caspase-3 in keratinocytes. This strongly suggests that RSR response driven by ZAKα is much broader than connecting to the sole NLRP1 inflammasome pathway, which highlights the relevance of detection of proper ribosomal function and protein translation for cellular homeostasis and survival. To this regard, the study of the P38 and JNK kinases, both activated by ZAKα and already described as strong inducers of Caspase-3-dependent cell death during RSR, will be of importance in order to discriminate about their respective importance in Portimine-driven skin inflammation (Sinha et al, 2024; Robinson et al, 2023). In addition, it is intriguing that Portimine A triggers such a strong inflammatory response in humans. This raises the key question of which species is the main target of Portimine A. Indeed, microalgae cohabit with a large variety of living organisms, which include multiple microorganisms, other algae, fishes as well as mammals (Abadie et al, 2018; Griffith and Gobler, 2020; Maggiore et al, 2020). Future studies regarding the ecological niches of *V. rugosum* during and before blooms will probably help answer this question.

Regarding the importance of RSR-driven hNLRP1 inflammasome, our findings agree with previous findings from different groups and our consortium on the identification of hNLRP1 as a central sensor of various threats-induced RSR response, which includes, but is not restricted to, UVB, crop- and microbial-associated toxins or several flaviviruses (Robinson et al, 2022; Pinilla et al, 2023; Robinson et al, 2023; Rodríguez-Ruiz et al, 2023; Zhou et al, 2023; Rozario et al, 2024; Burian and Yazdi, 2018; Jenster et al, 2023; Bauernfried et al, 2021). A key question regarding all those processes is whether the link between hNLRP1 and RSR has been selected by the human adaptation to environmental stressors (UVB exposure, crop/plants/algal ribotoxins), by a coevolution between host and microbes (Parameswaran et al, 2024) or by a yet-to-be-determined additional reason (Ball et al, 2022; Wang et al, 2023). Future epidemiological and evolutionary studies will probably shed light on this question. In addition, the hNLRP1 locus has been described to be highly polymorphic leading to the presence of numerous hNLRP1 variants (Vasseur et al, 2012). It will be extremely important to monitor and evaluate in the future if those variants are able to dampen/exacerbate NLRP1 responses as a result of higher UVB radiation or additional emerging ribotoxic threats caused by climate change. Keeping this in mind, our study provides some inputs by identifying two mutations in hNLRP1 that fail to trigger inflammasome assembly in response to Portimine A and potentially, to a greater extent, to various RSR inducers. Whether patients carrying those mutations exhibit a selective advantage to Portimine A and other RSRs will constitute an exciting field of investigations.

Our study does not directly address the molecular target of Portimine A upstream of its effect of ribosome stalling. This question was partially addressed by a recent study showing Portimine A directly binds to NMD3, an essential cytosolic protein required for the assembly of the cytosolic 60S ribosome (Tang et al, 2023). Curiously, knocking down NMD3 confers protection against Portimine A in cancer cell lines, rather than eliciting the same effects as Portimine A (Tang et al, 2023). These results suggest that Portimine A is unlikely to act directly as an inhibitor of NMD3. Future studies are required to clarify the mechanistic connection between Portimine A, NMD3 and RSR. Nonetheless, Portimine A might be a good candidate for anti-tumoral therapies due to its strong potency to kill tumoral cell lines but not primary immune cells (PBMCs) (Tang et al, 2023). In line with those suggestions, or results also support a model where naïve lymphocytes, monocytes and neutrophils from healthy donors are naturally resistant to Portimine A through a yet-to-be determined mechanism. However, our findings showing the extreme potency of

Portimine A at promoting RSR-dependent keratinocyte, airways and endothelial cell death at very low concentration (nM range) suggest that its use in a clinical setting as a chemotherapeutic agent requires further investigation. In its current form, Portimine A might have strong necrotic and inflammatory side effects on non-immune cells such as endothelial cells or airways/skin epithelia. Yet, this kind of highly cytotoxic compounds or vesicants, when formulated optimally for topical applications, are precisely what is required to treat hyperproliferative skin lesions such as benign acanthomas and viral warts (e.g., Aldara and Ycanth creams). Thus, we speculate that Portimine A could be formulated as a topical agent to induce ZAKα-dependent NLRP1 inflammasome activation in a delimited area of targeted tissue/cells.

Finally, although our study does not provide the final evidence that during the outbreaks in Cuba and in Senegal, Portimine A was directly responsible of the strong dermatitis observed, our results strongly point toward this direction and warrant for the necessity to address *V. rugosum* presence and concentration in various marine environments and compartments (including seawater and edible marine organisms) in order to prevent any adverse effects on human health.

Beyond our work on the ability of *V. rugosum* to induce PortimineA-dependent damages in humans, the aquatic world is currently subjected to intense remodeling due to human activity and environmental change. This ongoing global change leads, among others, to shifts in the distribution of microorganisms, with other toxic and epiphytic dinoflagellates being no exception to this trend. This change might lead to unpredictable adverse impact on human health, as exemplified by the outbreak of PortimineA associated dermatitis detailed in this study. This warrants for a general and massive effort to (1) address the presence of various metabolite or toxin-producing species, (2) understand their mechanism of action, and (3) find/develop novel solutions to predict and treat public health threats.

# Methods

### Reagents and tools table

| Reagent/resource | Reference or source | Identifier or catalog number |
|---|---|---|
| **Experimental models** | | |
| A549 ASC-GFP | Invivogen | a549-ascg |
| A549 ASC-GFP-NLRP1 | Invivogen | a549-ascgnlrp1 |
| HEK293-ASC-GFP | This study | This study |
| HEK293-ASC-GFP/NLRP10 | This study | This study |
| HEK293-ASC-GFP/NLRP1 | This study | This study |
| HEK293-ASC-GFP/NLRP3 | This study | This study |
| HEK293-ASC-GFP/AIM2 | This study | This study |
| NTERT NLRP1 KO + 86-275-SNAP | This study (franklin.zhong@ntu.edu.sg) | This study (franklin.zhong@ntu.edu.sg) |
| **Recombinant DNA** | | |
| pLvB72 hNLRP1 plasmid | Pinilla et al, 2023 | |
| pLvB72 hNLRP1 DelPro108_112T plasmid | Genscript | |
| pLvB72 hNLRP1 S107F plasmid | Genscript | |
| **Antibodies** | | |
| ZAK antibody 1:1000 | Bethyl Laboratories | A301-993A |
| P38 MAPK antibody 1:1000 | Cell Signaling | 9212S |

| Reagent/resource | Reference or source | Identifier or catalog number |
|---|---|---|
| Phospho-P38 MAPK (Thr180/Tyr182) (D3F9) 1:1000 | Cell Signaling | 4511S |
| Anti-NLRP1 (N-terminal) 1:500 | R&D Systems | AF6788 |
| Anti-Asc, pAb (AL177) 1:1000 (Immunoblotting) 1:250 (Immunofluorescences) | Coger | AG-25B-0006-C100 |
| Anti-puromycin, clone 12D10 1:1000 | Sigma-Aldrich | MABE343 |
| Anti-NMD3 Rabbit Polyclonal Antibody 1:1000 | Proteintech | 16060-1-AP |
| Anti-LSG1 Rabbit Polyclonal Antibody 1:1000 | Proteintech | 17750-1-AP |
| eIF6 Polyclonal Antibody 1:1000 | Invitrogen | 16548781 |
| Anti-RPL10 Rabbit Polyclonal Antibody 1:1000 | Proteintech | 17013-1-AP |
| Anti-α Tubulin antibody 1:10,000 | Abcam | ab4074 |
| Anti-β-actin 1:10,000 | Sigma-Aldrich | A1978 |
| IL-33 monoclonal antibody (Nessy-1) 1:1000 | Enzo Life Sciences | ALX-804-840B-C100 |
| Anti-GFP antibody 1:5000 | Abcam | Ab6673 |
| Goat-anti rabbit HRP secondary antibody 1:5000 | Advansta | R-05072-500 |
| Goat-anti mouse HRP secondary antibody 1:5000 | Advansta | R-05071-500 |
| **Oligonucleotides and other sequence-based reagents** | | |
| map3k20a: ENSDARG00000006978, Zaka: Zebrafish | | |
| zak-gRNA-2 5′ CTCTGTCCT GCGAGAGCCAG 3′ | Sigma-Aldrich | |
| zak-gRNA-3 5′ ACCTGCGAC ACCTTCTCCTA 3′ | Sigma-Aldrich | |
| zak-gRNA-4 5′ AAGCCCCTCCA GACCTTTGA 3′ | Sigma-Aldrich | |
| NLRP1: ENSDARG000000088423, ZDB-GENE-120709: Zebrafish | | |
| nlrp1-gRNA-2 5′ CCAGCTGACCA AGACTCCTG 3′ | Sigma-Aldrich | |
| nlrp1-gRNA-4 5′ TTGCTCCTCTG AATGATCAC 3′ | Sigma-Aldrich | |
| nlrp1-gRNA-5 5′ TGGTGCTGTGC TACTGTCTT 3′ | Sigma-Aldrich | |
| PYCARD: ENSDARG00000040076.8, ZDB-GENE-000511-2: Zebrafish | | |
| Pycard-gRNA-n°1 5′ CGTGTTCACATC AAAAGACG 3′ | Sigma-Aldrich | |
| Pycard-gRNA-n°3 5′ AAAGCAAACTGG GCGATCGG 3′ | Sigma-Aldrich | |
| Pycard-gRNA-n°4 5′ ATTGCAGACTTT GTGACGCG 3′ | Sigma-Aldrich | |
| HRM Sequencing primers for KO validation: | | |

| Reagent/resource | Reference or source | Identifier or catalog number |
|---|---|---|
| ZAKa-F2: TGTGTGTGTGT TTTGGAGCG | Sigma-Aldrich | |
| ZAKa-R2: TGTCTCCCACC TCTTTCTCGA | Sigma-Aldrich | |
| ZAKa-F3 AGATCTGTGA TTTCGGGGCG | Sigma-Aldrich | |
| ZAKa-R3 TGTGTCCTATG GTCAGATGTGA | Sigma-Aldrich | |
| ZAKa-F4 GCAGACATAG CTCCGGGTAC | Sigma-Aldrich | |
| ZAKa-R ATGTTTCTCC ACCACCAGCC | Sigma-Aldrich | |
| NLRP1 F2 FW CCATCCCAGA AAGCCCCAGTT | Sigma-Aldrich | |
| NLRP1 F2 RV CTCCTGCACCC TTCTCTCAGA | Sigma-Aldrich | |
| NLRP1 F4 FW GTCATGATGTT TCTCTGTGGCC | Sigma-Aldrich | |
| NLRP1 F4 RV CGTGACAAATC TCTCATCCTGGA | Sigma-Aldrich | |
| NLRP1 F5 FW CACGAAAGTTC TGGAAACACACA | Sigma-Aldrich | |
| NLRP1 F5 RV ATTTGACCAATA TAAAGAAACATTAACAGA | Sigma-Aldrich | |
| PYCARD n°1,3,4 FW GAACCATGTAGC GGAATCTTTC | Sigma-Aldrich | |
| PYCARD n°1,3,4 RV GCCCTGTGTTCCT CAATAGATCA | Sigma-Aldrich | |
| PYCARD n°1,3,4 FW GAACCATGTAGCG GAATCTTTC | Sigma-Aldrich | |
| PYCARD n°1,3,4 RV GCCCTGTGTTCCT CAATAGATCA | Sigma-Aldrich | |
| PYCARD n°1,3,4 FW GAACCATGTAGCG GAATCTTTC | Sigma-Aldrich | |
| PYCARD n°1,3,4 RV GCCCTGTGTTCCTC AATAGATCA | Sigma-Aldrich | |
| **Chemicals, enzymes, and other reagents** | | |
| Purified Portimine A 4.78 ng/mL | Novakits | STD-PORT |
| Purified Portimine B 40 or 400 ng/mL | Novakits | STD-PORTB |
| CRM- Pinnatoxin G/H 40 ng/mL | Novakits | NRC-CRM-PNTX-G |
| Sample II 25 µg/mL from extract (corresponding to 6,249 ng/mL of Portimine A) | IFREMER/IRD | N.A. (This study) |
| Anisomycin 1 µM | Selleck | SE-S7409-10 MG |
| PhoSTOP | Sigma-Aldrich/Roche | 4906845001 |
| cOmplete, Mini, EDTA-free protease inhibitor cocktail | Sigma-Aldrich/Roche | 4693159001 |
| Lipofectamine LTX | Invitrogen | 15338030 |

| Reagent/resource | Reference or source | Identifier or catalog number |
|---|---|---|
| Molecular probes SYTOX Green nucleic acid stain | Invitrogen | S7020 |
| Nate | Invivogen | lyec-nate |
| Phos-tag Acrylamide | Wako Chemicals | AAL-107 |
| Manganese chloride (II) | Sigma-Aldrich | 63535 |
| Prestained protein size marker III | Wako Chemicals | 230-02461 |
| DMEM | Gibco | 41965039 |
| Opti-MEM | Gibco | 31985047 |
| Trypsin-EDTA | Gibco | 25200056 |
| TrypLE | Gibco | 12604013 |
| Keratinocyte Growth Medium 2 | PromoCell Inc. | C-20011 |
| Endothelial Cell Growth Medium MV | PromoCell Inc. | C-22020 |
| b-Raf/ZAK inhibitor (PLX4720) 10 µM | MedChem Express | HY-51424 |
| SB 203580 10 µM | MedChem Express | HY 10256 |
| Doramapimod 10 µM | MedChem Express | HY-10320 |
| Z-YVAD 50 µM | Invivogen | inh-yvad |
| Z-DEVD 20 µM | Selleck Chemicals | S7312 |
| Z-VAD 10-100 µM | Invivogen | vad-tlrl |
| Cycloheximide | Sigma-Aldrich | C4859-1ML |
| 6p 1 µM | Yang et al, 2020 | Yang et al, 2020 |
| Emricasan 5 µM | MedChem Express (MCE) | HY-10396 |
| SNAP-Cell TMR-Star | NEB | S9105S |
| Human IL-1 Family Cytokine Array C1 | Tebu-bio | AAH-IL1F-1-8 |
| Human Inflammation Antibody Array | Abcam | ab134003 |
| IL-1 β Human Uncoated ELISA Kit | Invitrogen | 88-7261-77 |
| Human Total IL-18 DuoSet ELISA | Bio-Techne | DY318-05 |
| Human IL-1α Platinum ELISA Kit | Invitrogen | BMS243-2 |
| Cyquant LDH | Invitrogen | C20301 |
| MACSxpress Whole Blood Neutrophil Isolation Kit, human | Miltenyi Biotec | 130-104-434 |
| MACSxpress Buffy Coat CD4 T Cell Isolation Kit, human | Miltenyi Biotec | 130-120-003 |
| MACSxpress Whole Blood monocyte Isolation Kit, human | Miltenyi Biotec | 130-093-545 |
| TNT T7 Coupled Reticulocyte Lysate System | Promega | L4610 |
| **Software** | | |
| Prism10.2.3 | N.A. | GraphPad |
| Biorender | N.A. | Biorender.com |
| **Other** | | |

## Use of human cells and animal agreements

Zebrafish husbandry and experiments were conducted in compliance with guidelines from the Spanish (RD 53/2013, RD 1386/2018 and Law 6/2013) and European (2010/63/EU and ECC/566/2015) Legislation and approved by the Animal Experimentation Ethics Committee (CEEA,

University of Murcia) and *Consejeria de Agua, Agricultura, Ganaderia y Pesca, Región de Murcia*, references A13230902 and A13240102.

All primary keratinocyte and 3D organotypic skin experiments were carried out with approval from the Agency for Science, Technology and Research Human Biomedical Research Office (A*STAR Full IRB-2020-209). For the 3D organotypic cultures, primary human keratinocytes and fibroblasts were obtained from a single donor aged 49, female of Chinese ethnicity.

Human whole blood cell use from healthy donors was carried out in under approval from the "Etablissement Français du Sang" (EFS, Toulouse, France) and the CNRS (agreement 21PLER2020-025).

Informed consent was obtained from all subjects and the experiments conformed to the principles set out in the WMA Declaration of Helsinki and the Department of Health and Human Services Belmont Report.

All cells were authenticated and regularly tested for mycoplasma contamination with the Mycostrip assay (rep-mysnc-100, Invivogen).

## Reagents

All reagents used in this study as well as their concentration of use and origin are listed in the Reagent Tools Table.

## Environmental compartment sampling during the outbreaks

Sampling of environmental compartments, including biomass from the fishing canoe (I), biomass from the fishing drift-net (II), GF/F filtered seawater (III), sediment (IV), and mussel flesh (V), was conducted during both outbreaks in 2020 and 2021 at different locations where the contaminated fishermen operated (Fig. 1A–C).

## Microscopy

For light microscopy, cells were isolated with a micropipette using an IX51 inverted microscope (Olympus, Tokyo, Japan). They were transferred in a drop of clean water on a slide and covered with a 0.17-mm-thick coverslip. Photomicrographs were taken with a Zeiss Universal microscope fitted with an EOS-M (Canon, Tokyo, Japan) digital camera. To better visualize thecal plates, microscopy was performed by adding a drop of Solophenyl Flavine 7GFE 500 (Ciba Specialty Chemicals, High Point, North Carolina USA).

## Culture of *Vulcanodinium rugosum*

*V. rugosum* strain IFR-VRU-01, originally isolated from the Ingril Lagoon, France (WoRMS—World Register of Marine Species—*Vulcanodinium rugosum* Nézan & Chomérat, 2011), was used as organism model for this study. It was cultivated in L1 growth medium (Guillard, 1975). The culture medium was prepared using filter-sterilized (0.2 μm) Mediterranean seawater (38 psu). The culture was maintained at 25 °C, with a photon flux density of ca. 100 μmol m$^{-2}$ s$^{-1}$, under a 12:12 h light/dark photoperiod. Cells were harvested during the stationary growth phase by centrifugation. Finally, the wet algal biomass obtained was freeze-dried and ground to a fine powder.

## DNA extraction and qPCR assay

The Sample I (Canoe Biomass), from 2021, and one strain of *Vulcanodinum rugosum* (IFR-CC 19-082) isolated from the type

locality (Ingril Lagoon, France) were extracted using the DNeasy PowerLyzer PowerSoil Kit (Qiagen) with MN Bead Tubes Type C (Macherey-Nagel, Germany) and diluted in 50 μL of C6 buffer. DNA extracts were quantified with the dsDNA HS Assay Kit (Invitrogen). Quantitative PCR analysis, based on the protocol described by (Bouquet et al, 2023a), was performed in 96-well plates using a QuantStudio 5 Real-Time PCR system (Applied Biosystems). The Platinum SYBR Green qPCR SuperMix-UDG kit (Invitrogen) was used. Each reaction had a final volume of 25 μL, consisting of 12.5 μL of SuperMix, 0.5 μL of each primer (VulcaF/VulcaR, 200 nM final concentration), 10.5 μL of Nuclease-Free Water, and 1 μL of DNA template. The quantification cycling protocol was as follows: an initial denaturation step at 95 °C for 2 min, followed by 40 cycles of 15 s at 95 °C and 30 s at 60 °C. The melting curve profile was generated by increasing the temperature from 60 to 95 °C at a rate of 0.15 °C per second. The amplified region was 132 bp.

## Sample preparation and extraction

Different extraction processes were employed depending on the samples. The lyophilized biomass (20 mg) from the cultivated *V. rugosum* was extracted twice with 1 mL of methanol (MeOH, ≥99.9%, CHROMASOLV, Honeywell Riedel-de Haën), with vortexing and ultrasonication in an ice bath (25 kHz, sweep mode, 15 min). Cell pellets were removed from the extract by centrifugation (3500 × g, 15 min, 4 °C). For biomass from the canoe bottom (I), or from the fishing net (II), and sediment (III), each lyophilized sample (50 mg) was combined with 500 mg of glass beads (0.15–0.25 mm; VWR) and 2 mL of MeOH (Honeywell) before mechanical grinding using a mixer mill (Mixer Mill MM400, Retsch) for 20 min at 30 Hz. After centrifugation (4300 × g, 10 min, 4 °C), the pellets were subjected to an additional extraction with 2 mL of methanol. Finally, the two supernatants were pooled, evaporated to dryness with a gentle flow of nitrogen at 40 °C, and resuspended in 1 mL of methanol. GF/F filters (IV) were extracted similarly, except they were re-extracted twice with 2 mL MeOH and resuspended in 0.75 mL after evaporation. Mussel extracts (V) were prepared by vortexing 2 g of homogenized fresh tissue with 9 mL of methanol, followed by extraction using an ultrasonic bath (25 Hz, sweep mode, 15 min, Brandsonic). After centrifugation (4300 × g, 10 min, 4 °C), the supernatant was transferred to 20-mL volumetric flasks. This extraction process was repeated, and the volumetric flask was filled to 20 mL with MeOH. All resulting extracts were ultrafiltered (3500 × g, 30 s, 4 °C, 0.2-μm Nanosep filters) and stored at −20 °C until analysis.

The samples were analyzed on a UFLC (Shimadzu) coupled to a triple-quadrupole mass spectrometer (API4000Qtrap, Sciex) equipped with a heated electrospray ionization (ESI) source. Instrument control, data processing, and analysis were conducted using Analyst software 1.7.2. Chromatography was performed using a Kinetex C18 or XB C18 column (100 × 2.1 mm, 2.6 μm) with a suited guard column. A binary mobile phase was used, phase A (100% aqueous) and phase B (95% aqueous acetonitrile or 95% aqueous MeOH), both containing 2 mM ammonium formate and 50 mM formic acid. The flow rate was 0.3 mL.min$^{-1}$, and the injection volume was 5 μL. The column and sample temperatures were 40 °C and 4 °C, respectively. Gradient elution was employed, starting with 5% B, rising to 90% B over 12 min, held for 3 min,

then decreased to 5% B in 0.1 min and held for 3.0 min to equilibrate the system.

The ESI interface was operated using the following parameters: curtain gas 25 psi, temperature: 550 °C, gas1 50 psi; gas2 55 psi, ion spray voltage 2000 V. The dwell time was 20 ms. The transitions and MS/MS parameters used for the MRM (Multiple Reaction Monitoring) mode in positive ionization are reported in the "Plessis L et al, 2024 (LC-MS/MS raw data for Fig. 1C [Data set])".

The certified reference standard of PnTX-G was purchased from the National Research Council Canada (NRCCNRC, Halifax, Canada), and the non-certified standards (PnTX-A, PnTX-E, PnTX-F, PnTX-H, Portimine A, and Portimine B) were purchased from the Cawthron Institute, New Zealand. Quantification was performed using 7-point linear calibration curves generated from reference standards of PnTX-A, PnTX-E, PnTX-F, PnTX-G, PnTX-H, and Portimine A. The limit of quantification (LOQ) for the standards was 0.15 ng mL$^{-1}$, and the limit of detection (LOD) was 0.05 ng mL$^{-1}$.

## Zebrafish husbandry

Zebrafish AB wild-type strain (Zebrafish International Resource Center) (ZFIN ID: ZDB-GENO-960809-7) was used. Adult zebrafish were maintained in buffered reverse osmotic water systems and were exposed to a 14-h light–10-h dark cycle to maintain proper circadian conditions (Roca et al, 2019). Zebrafish embryos were housed at 28.5 °C in fish water (reverse osmosis water containing 0.18 g/l Instant Ocean supplemented with 0.25 mg/mL methylene blue) from collection to 1 day post-fertilization (dpf) and in 0.5× E2 Embryo Medium (E2/2) supplemented with 0.003% 1-phenyl-2-thiourea (PTU) (Sigma) from 1 dpf to prevent pigmentation (Roca et al, 2019).

## PortimineA administration to zebrafish larvae and assessment of survival, gross caudal fin pathology, and cell necrosis

Portimine A, at the concentrations indicated, was administered once to larvae (of undetermined sex given the early developmental stages used) at 2 dpf and kept for the duration of the experiment. Dechorionated sibling larvae were mixed in a Petri dish and held at 28.5 °C before random allocation to the PortimineA-treated or control groups; 0.021–0.42% methanol (Sigma) (PortimineA carrier, maximum dose used for all experimental groups in the same experiment) was used as the control (vehicle). All groups were maintained in 0.5% DMSO (Sigma) to increase permeability. Sample size was determined based on pilot experiments. For survival experiments, zebrafish larvae were checked twice daily, and skin damage and swim-away touch-induced reflex assessed; any animal showing severe skin damage and not responding to three touches with a platinum wire probe to test the swim-away reflex was euthanized immediately and recorded as dead in the survival experiment. Survival and caudal fin gross pathology were assessed using a Nikon SMZ800N dissecting microscope and images were taken using a camera Sony IMX290C with full HD resolution camera. Fluorescence microscopy was performed as described (Takaki et al, 2013). We assessed cell necrosis in the caudal fin of the larvae by using SYTOX Green staining. PortimineA-treated and control zebrafish larvae were incubated in 400 nM SYTOX Green in

E2/2 medium for 10 min, then washed twice, anesthetized in 0.025% Tricaine (Sigma), and embedded in low-melting-point agarose as previously described (Roca et al, 2019). Skin damage severity was assessed in the caudal fin of each larva using two methods: assessing the damage as no damage or low, moderate, or severe damage and quantifying the area of the tail in each fish as previously described (Roca et al, 2019). Brightfield or fluorescence images were taken with NIS Elements (Nikon) using a Nikon Eclipse Ti2-E microscope fitted with Nikon Plan Fluor 4 × 0.2 NA and Nikon Plan Fluor 10 × 0.45 NA objectives.

## Generation of ZAKa-, NLRP1- and Asc-deficient zebrafish larvae

G0 Zaka-, Nlrp1-, and Asc-deficient zebrafish embryos (crispants) were generated using CRISPR-Cas9 technology by simultaneously targeting different sites of the genes of interest (Wu et al, 2018) (Fig. EV4B). This methodology is used to create chimeric animals with different degrees of protein knockdown (Wu et al, 2018). Guide RNAs were prepared following the manufacturer specifications, hybridizing the common RNA component (Alt-R tracrRNA) with each of the specific Alt-R crRNA. In total, 3–5 nl of a solution containing Alt-R crRNA and Alt-R tracrRNA (30 µM each) complexed with Cas9 protein (0.25 µg/µl) (Integrated DNA Technologies), and 2% phenol red sodium salt (Sigma) was injected into 1–2 cell stage embryos (Wu et al, 2018). Similar volumes of a solution containing Cas9 protein and phenol red were used to generate the control animals. The genotype of individual larvae and mutagenesis efficacy were assessed by high-resolution melt (HRM) analysis (Garritano et al, 2009) using the guides and primers described in the Reagent Tools Table.

## Assessment of neutrophil recruitment in zebrafish larvae

Neutrophil recruitment was assessed using the zebrafish line Tg(LysC:GFP)^nz117 (with GFP-expressing neutrophils) embedded in low-melting point agarose as described before. Neutrophils recruited to the damaged area of the tail were counted from the end of the caudal aorta to the edge of the caudal fin, as illustrated in (Fig. 4G).

## Cell culture

HEK293 and A549 cells were maintained in Dulbecco's modified Eagle's medium (DMEM; Gibco) supplemented with 10% FCS and 1% penicillin–streptomycin at 37 °C 5% $CO_2$.

Immortalized N/TERT keratinocytes were provided by J. Rheinwald (Material Transfer Agreements to FL Zhong and E. Meunier). WT, ZAKα KO, NLRP1 KO cells were previously described by (Zhong et al, 2018; Robinson et al, 2022) and NLRP1 KO cells complemented with NLRP1 Disordered Region (DR) construct (aa 86-275) expressing a SNAP tag (here referred to as NTERT NLRP1 KO + 86-275-SNAP) were generated in the frame of this study.

Primary Human Keratinocytes (pHEKs, PromoCell) and NHEK-Neo (Neonatal Norman Human Epidermal Keratinocytes, 00192907, Lonza, Switzerland) were maintained in Keratinocyte Growth Medium 2 (PromoCell) at 37 °C 5% $CO_2$. Primary Human Endothelial Cells (pHECs) were maintained in Endothelial Cell Growth Medium MV (PromoCell Inc) at 37 °C 5% $CO_2$.

Primary Human Nasal Epithelial Cells (pHNECs) were collected on superior turbinates using smear brushes at the Hospital of Toulouse, France as previously described. Briefly, basal cells were counted and seeded onto collagen-coated (0.03 mg/mL) and maintained in Pneumacult Ex Plus Medium (StemCell) at 37 °C 5% $CO_2$.

Whole blood was collected from healthy donors by the "Etablissement Français du Sang" (EFS, Toulouse, France). Neutrophils were then isolated by negative selection using MACSxpress Whole Blood Human Neutrophil Isolation Kit (Miltenyi Biotech) according to the manufacturer's instructions. CD4 T and monocyte cells were isolated by positive selection using MACSxpress Buffy Coat CD4 T and monocyte Cell Isolation Kit according to the manufacturer's instructions.

## Cell stimulation

Otherwise specified, cells were plated 1 day before stimulation in six-well plates at $2 \times 10^5$ cells per well; 12-well plates at $1 \times 10^5$ cells per well or 96-well plates at $2 \times 10^4$ per well in 1 mL of DMEM, 10% FCS, and 1% penicillin/streptomycin or in Keratinocyte Growth Medium 2 (PromoCell Inc).

Next day, medium was changed to OPTI-MEM and cells were preincubated or not with the indicated inhibitors for 1 h (h). All cells were treated with the indicated concentration of Sample II (25 µg/mL from extract (corresponding to 6249 ng/mL of Portimine A)), Portimine A (4 ng/mL), Portimine B (400 ng/mL), PnTX-G (40 ng/mL), and anisomycin (1 µM) for indicated times.

## Cytotoxicity (MTT assay)

Cells were seeded in a 96-well tissue culture-treated plate (734-4058, VWR, France) at a density of 20,000 cells per well in 200 µL of medium without hydrocortisone and incubated at 37 °C, 5% $CO_2$ for 3 h to ensure cell adhesion. Exposure was performed by adding 6 µL of standard toxins or sample extract dissolved in MeOH (< 3% of total well volume) in a 9-point serial dilution. Blank methanol and untreated cells were included as controls. Each condition was set in triplicate ($n = 3$) on the plate. Cell viability was assessed on the following day using the MTT colorimetric assay. After removing the culture medium, 50 µL of MTT solution (0.8 mg mL$^{-1}$ in PBS) were added to each well and incubated for 3 h before measurement. Following incubation, the MTT solution was removed, and the metabolized formazan dye was solubilized in 100 µL of DMSO. The optical density was then measured at 540 nm using a microplate reader (CLARIOstar PLUS, BMG Labtech, France). GraphPad Prism software (version 10.1.2) was used to plot sigmoid curves from at least three independent experiments. The half maximal inhibitory concentration ($IC_{50}$) was determined using nonlinear regression analysis (curve fitting) with the "[Inhibitor] vs. response—Variable slope (four parameters)" model.

## CRISPR-Cas9 ribonucleoprotein genome editing in primary keratinocytes

To form the CRISPR-Cas9 ribonucleoprotein (cRNP), two ZAKa crRNAs, 1 µL each (20 µM; IDT) were mixed with 2 µL tracrRNA (20 µM; #1072532; IDT) and 1 µL duplex buffer (IDT). The following guide sequences were used to edit the ZAKa gene (5′-ATTCTTGAACCTCCCAACTA-3′ FWD; 5′-GTGACAATGCCA TAGTTGGG-3′ REV). The RNAs were then annealed together by heating to 95 °C, before ramping down the temperature by 5 °C per minute to 20 °C. To assemble the cRNP 4 µL of the crRNA-tracrRNA duplex was mixed with 1.5 µL Cas9 nuclease (#1081058; IDT) and 2 µL PBS. The resulting mixture was left at room temperature for a minimum of 20 min.

The 7.5 µL cRNP was then mixed with $6.5 \times 10^5$ primary keratinocytes in electroporation buffer comprised of 17 µL P3 and 3.6 µL supplement (#V4XP-3032; Lonza). The final mixture was applied to the cassette and electroporated in an Amaxa Nucleofector (#AAF-1003X; Lonza), using code DS138. Primary keratinocytes were then expanded for validation and banking. Validation of CRISPR-Cas9 gene editing of ZAKa was done by western blot (250 ng/mL; #A301-993A).

## Organotypic cultures, stimulation, and analysis

Skin organoid cultures were generated by adapting a previously described protocol (Arnette et al, 2016). Briefly, 2 mL of collagen I (4 mg/mL; #354249; Corning) mixed with $7.5 \times 10^5$ human fibroblasts were allowed to polymerize over 1 mL of acellular collagen I in six-well culture inserts (#353102; Falcon) placed in 6-well deep well plates (#355467; Falcon). After 24 h, $1 \times 10^6$ primary human keratinocytes were seeded into the inserts and kept submerged in a 3:1 DMEM (#SH30243.01; Hyclone) and F12 (#31765035; Gibco) mixture with 10% FBS (#SV30160.03; Hyclone), 100 U/mL of penicillin–streptomycin (#15140122; Gibco), 10 µM Y-27632 (#1254; Tocris), 10 ng/mL of EGF (#E9644; Sigma-Aldrich), 100 pM cholera toxin (#BML-G117-001; Enzo), 0.4 µg/mL of hydrocortisone (#H0888; Sigma-Aldrich), 0.0243 mg/mL adenine (#A2786; Sigma-Aldrich), 5 µg/mL of insulin (#I2643; Sigma-Aldrich), 5 µg/mL of transferrin (#T2036; Sigma-Aldrich), and 2 nM 3,3′,5′-triiodo-L-thyronine (#T6397; Sigma-Aldrich). After another 24 h, the organotypic cultures were then raised at the air–liquid interface and fed with the submerged media (without Y-27632 and EGF) below the insert to induce epidermal differentiation. The airlifting medium was replaced every 2 days, at 12 days post airlifting 20 µM Portimine A was added into the media until day 14 post airlifting. Organotypic treatments were performed as three technical replicates. Organotypic cultures were then harvested after treatment and formalin-fixed for 24 h. Fixed tissues were then embedded into wax for histological purposes before being cut and stained using a standard H&E protocol.

## Quantification of epithelial damage in 3D organotypics

H&E images from three individual sections were captured for each 3D organotypic technical replicate within the same experiment. To quantify intraepidermal disruptions, the total epidermis area was manually demarcated in Adobe Photoshop and its area was determined, after which all the intraepidermal disruptions were demarcated, and their total area was found. From these two metrics, a percentage of disruption/epidermis area was derived, these analyses were performed on three technical replicates per treatment.

## Soluble mediator release analysis

Human IL-1β enzyme-linked immunosorbent assay (ELISA) kit (#557953; BD) was used according to the manufacturer's protocol.

Culture supernatants of 3D skin were also collected and sent for Luminex analysis using the ProcartaPlex, Human Customized 65-plex Panel (Thermo Fisher Scientific) and analyzed for the following targets:

G-CSF (CSF-3), GM-CSF, IFN alpha, IFN gamma, IL-1 alpha, IL-1 beta, IL-2, IL-3, IL-4, IL-5, IL-6, IL-7, IL-8 (CXCL8), IL-9, IL-10, IL-12p70, IL-13, IL-15, IL-16, IL-17A (CTLA-8), IL-18, IL-20, IL-21, IL-22, IL-23, IL-27, IL-31, LIF, M-CSF, MIF, TNF alpha, TNF beta, TSLP, BLC (CXCL13), ENA-78 (CXCL5), Eotaxin (CCL11), Eotaxin-2 (CCL24), Eotaxin-3 (CCL26), Fractalkine (CX3CL1), Gro-alpha (CXCL1), IP-10 (CXCL10), I-TAC (CXCL11), MCP-1 (CCL2), MCP-2 (CCL8), MCP-3 (CCL7), MDC (CCL22), MIG (CXCL9), MIP-1 alpha (CCL3), MIP-1 beta (CCL4), MIP-3 alpha (CCL20), SDF-1 alpha (CXCL12), FGF-2, HGF, MMP-1, NGF beta, SCF, VEGF-A, APRIL, BAFF, CD30, CD40L (CD154), IL-2R (CD25), TNF-RII, TRAIL (CD253), TWEAK.

## Plasmid and cell transfection

Cells were plated in a six-well plate at $2 \times 10^5$ cells per well in 1 mL of DMEM complete medium. The following day, cells were incubated with Nate $1\times$ (Invivogen) for 30 min. In all, 1 µg of previously described NLRP1 plasmids (pLvB72 hNLRP1) (Pinilla et al, 2023) were transfected using lipofectamine LTX and PLUS reagent according to the manufacturer's instructions (Invitrogen).

Additional NLRP1 mutations (DelPro108_112T and S107F) (NLRP1 | gnomAD v4.1.0 | gnomAD) were further generated by Genscript in pLvB72 hNLRP1 by using site-directed mutagenesis. Transfected cells were incubated for 24 h before any further stimulation or not.

## Genetic invalidation using CRISPR-Cas9

Genetic invalidation of ZAKα and NLRP1 genes in pHEKs were achieved by using the previously described strategy (Pinilla et al, 2023). Briefly, LentiCRISPR-V2 vectors containing sgRNA guides against ZAKa and NLRP1 (ZAKα (MAP3K20)5′-TGTATGGTT ATGGAACCGAG-3′ FWD; NLRP1 5′-GATAGCCCGAGTG-CATCGG-3′ FWD) were used to transduce pHEKs. Transduced pHEKs were then selected with 1 µg/mL of Puromycin for 2 days and left recovering for 4 additional days before using them for experiments. Genetic invalidation efficiency was checked by Immunoblot on the whole cell population after antibiotic selection.

The use of A549 cells KO for ZAKα has been previously described (Pinilla et al, 2023).

## ASC speck imaging

ASC speck formation in A549 or HEK293 cells were monitored using an EVOS 7000 fluorescence microscope using a ×10 or ×20 objective. Quantification and analysis of ASC specks were performed by determining the ratios of ASC aggregates (i.e., ASC speck) formed in each cell over the total nuclei numbers (staining with Hoechst 33342) by using EVOS Analysis software. Quantifications were performed in a blinded way over three independent wells with a minimal number of cells of 500/well.

## Cell lysis assays

Cell lysis was evaluated by the quantification of LDH release into the cell supernatant, employing the LDH CyQUANT kit from

Thermo Fisher Scientific. pHEK were seeded in 96-well plates at $2 \times 10^4$ cells per well in Keratinocyte Growth Medium 2 (PromoCell Inc). The following day, cells were stimulated and 24 h after, 50 µL of cell supernatant was mixed with an equal volume of LDH substrate and left to incubate for 30 min at room temperature, protected from light. The enzymatic reaction was stopped by adding 50 µL of stop solution. Maximal cell death was determined with whole cell lysates from unstimulated cells incubated with 1% Triton X-100.

## Plasma membrane permeabilization monitoring

Indicated cells were plated at density of $2 \times 10^4$ per well in Black/Clear 96-well Plates in OPTI-MEM culture medium supplemented with SYTOX-Green dye (500 nM) and treated as mentioned in the figure legend. Green fluorescence was measured in real-time using Clariostar plate reader equipped with a 37 °C cell incubator. Maximal plasma membrane permeabilization levels were determined with whole cell lysates from unstimulated cells incubated with 1% Triton X-100.

## Cytokine quantification

Cells were seeded in 96-well plates at $2 \times 10^4$ cells per well and stimulated as indicated. Cytokine release was quantified 24 h after stimulation by ELISA kits according to the manufacturer's instructions for IL-1α, IL-1β, and IL-18.

For the Human Family Cytokine Array (listed in Reagent Table and Figs. 1 and EV1), pHEKs were seeded in T-75 flask at $1 \times 10^6$ cells in Keratinocyte Growth Medium 2 (PromoCell Inc). The following day, after stimulation, cell supernatant was collected and cytokine secretion was quantified according to the manufacturer's instructions.

## Immunoblot

Cells were seeded in six-well plates at $2 \times 10^5$ cells per well in DMEM, 10% FCS, 1% penicillin/streptomycin. The following day, after stimulation, cells were lysed in radioimmunoprecipitation assay (RIPA) buffer (150 mM NaCl, 50 mM Tris-HCl, 1% Triton X-100, 0.5% Na-deoxycholate) supplemented with protease inhibitor cocktail (Roche). Cell lysates were separated by denaturing SDS-PAGE and transferred to the polyvinylidene fluoride membrane. After transfer, the membrane was saturated for 1 h at room temperature in TBS-T (Tris 10 mM, pH 8, NaCl 150 mM, Tween 20 0.05%) containing 5% BSA or milk. Then, membranes were incubated overnight at 4 °C with the different primary antibodies. The membranes were then washed three times with TBS-T and incubated for 1 h at room temperature with secondary HRP-conjugated antibodies diluted in TBS-T. Then, the signal was detected with ECL revelation kit (Advansta) on a C-DiGit Imaging System (Li-cor). The primary antibodies and secondary antibodies used are listed in the Reagent Tools Table.

## Phosphoblots

Cells were seeded in six-well plates at $2 \times 10^5$ cells per well in DMEM 10% FCS, 1% penicillin/streptomycin. The following day,

after stimulation, cells were lysed in RIPA buffer (150 mM NaCl, 50 mM Tris-HCl, 1% Triton X-100, 0.5% Na-deoxycholate) supplemented with protease inhibitor cocktail (Roche) and phosphatase inhibitors cocktails (Roche). The collected cell lysate was supplemented with LaemLi buffer before boiling for 10 min at 95 °C. Cell lysates were then separated by SDS-PAGE and handled as described in "Immunoblot".

## PhosTag SDS-PAGE

Cells were seeded in 6-well plates at $2 \times 10^5$ cells per well in DMEM 10% FCS, 1% penicillin/streptomycin. The following day, after stimulation, cells were lysed in RIPA buffer supplemented with LaemLi buffer and protease and phosphatase inhibitor cocktails. Samples were then boiled for 10 min at 95 °C. Cell lysates were separated by PhosTag SDS-PAGE and the size of proteins was determined by the following size marker: wide-view prestained protein size marker III (Wako Chemicals). Briefly, 30 μM Phos-tag Acrylamide (AAL-107; Wako Chemicals) and 60 μM MnCl2 (63535; Sigma-Aldrich) were added to homemade 10% SDS-PAGE gel. Once the run was completed, the polyacrylamide gel was washed in transfer buffer with 10 mM EDTA twice, subsequently washed without EDTA twice, transferred to polyvinylidene fluoride membranes, thanks to a Trans-Blot Turbo (Bio-rad), blocked with 5% milk in TBS-T, and incubated with primary and corresponding secondary antibodies.

Specific to NLRP1 phosphorylation studies, N/TERT cells expressing the aa 86-275 of NLRP1 tagged with SNAP were seeded at a density of $8.10^4$ cells per well in 24-well plates a day prior. Cells were pre-treated with pan-caspase inhibitor Emricasan at a final concentration of 5 μM to avoid lytic cell death and treated with various combinations of triggers and inhibitors. SNAP-Cell TMR-Star was added to each well at a final concentration of 0.1 μM in the final hour of treatment to label the SNAP-tagged proteins. Cells were harvested directly in 1× Laemmli buffer and PhosTag gel electrophoresis was performed followed by direct gel visualization with a ChemiDoc MP (Bio-rad) using a standard rhodamine filter to image the fluorescently labeled SNAP-tagged proteins.

## Puromycin incorporation assays

Global translation rates were estimated based on the incorporation of puromycin. Cells were seeded in 12-well plates at $1 \times 10^5$ cells per well in DMEM 10% FCS, 1% penicillin/streptomycin. The following day, cells were stimulated with Sample, Portimine A (4 ng/mL), Portimine B (400 ng/mL), and PnTX-G (40 ng/mL) for indicated times. 30 min before the end of the stimulation, puromycin antibiotic was added in cell medium at 1 μg/mL final. Following puromycin incubation, cells were prepared for immunoblot. Puromycin incorporation was revealed using the antibody anti-puromycin antibody (clone 12D10 MABE343; Sigma-Aldrich).

## In vitro translation assay

The in vitro translation assay was performed by using rabbit reticulocytes lysates with the TNT T7 Coupled Reticulocyte Lysate System (L4610; Promega) following the manufacturer's recommendations. Translation was measured by the production of a reporter protein, here human Interleukin (IL)-33. The expression of the IL-33 was visualized by western blot.

## Polysomes profiling

Sucrose-gradient preparation: Five sucrose solutions containing 10%, 20%, 30%, 40%, and 50% sucrose (wt/vol) were prepared in TMK buffer (20 mM Tris-HCl, pH 7.4, 10 mM $MgCl_2$, 50 mM KCl). Layers of 2.1 mL of each solution were successively poured into 12.5 mL polyallomer tubes (Cat. # 331372; Beckman-Coulter) starting from the most concentrated solution (50%) at the bottom of the tubes to the least concentrated solution (10%) at the top. Each layer was frozen in liquid nitrogen before pouring the following one. Frozen gradients were stored at −80 °C and slowly thawed overnight at 4 °C before use. For extract preparation, A549, HEK293 and pHEKs cells were grown to 70% confluency and then treated or not with Portimine A (4 ng/mL) at the indicated times. Following this treatment, cells were incubated with cycloheximide (CHX, 100 μg/mL) for 15 min at 37 °C. Then, cells were rinsed twice with PBS and treated with 1 mL of trypsin (0.25%) for 5 min at 37 °C. Trypsin was diluted with DMEM medium containing 100 μg/mL of CHX. Cells were mixed up and down and counted to adjust the final resuspension volume in each condition. Cells were centrifuged at 1200 rpm ($300 \times g$) for 5 min at 4 °C. The cell pellet was washed with ice-cold PBS containing 100 μg/mL of CHX. Cells were centrifuged at 1200 rpm ($300 \times g$) for 5 min at 4 °C. Supernatants were aspirated and gently lysed in lysis buffer (20 mM Tris-Cl [pH 8], 150 mM KCl, 15 mM $MgCl_2$, 1% Triton X-100, 1 mM dithiothreitol, 100 μg/mL CHX, EDTA-free and protease inhibitor cocktail). Samples were incubated on ice for 20 min and centrifuged at $1000 \times g$ for 5 min at 4 °C min and supernatant corresponding to the cytosolic fraction of the cells was collected into a 1.5-mL tube. Samples were further centrifugated at $10,000 \times g$ for 5 min at 4 °C to clarify the cytoplasmic extract. Extracts were quantified by measuring absorbance at 260 nm using Nanodrop.

## Extracts loading, gradient centrifugation, and collection

Normalized amounts of extracts were loaded on 10–50% sucrose gradients and then centrifuged at $260,800 \times g$ for 2.5 h at 4 °C in an Optima L-100XP ultracentrifuge (Beckman-Coulter) using the SW41Ti rotor with brake. Following centrifugation, fractions were collected using a Foxy R1 gradient collector (Teledyne Isco) driven by PeakTrak software (Version 1.10; Isco Inc.). $A_{254}$ was measured during collection with a UA-6 UV/VIS DETECTOR (Teledyne Isco). The final polysome profiles were generated in Excel from. txt files extracted from PeakTrak software.

## Statistical analysis

All important information regarding the statistical analysis test used is included in figure legends. We used Prism (GraphPad Software, Inc.) to perform statistical analysis. Otherwise described, data are reported as mean with SEM. When comparing two groups, one-way ANOVA was chosen and multiple group comparisons were analyzed by using two-way ANOVA with multiple comparisons test. For survival analyses, Log-rank (Mantel–Cox) test and

**The paper explained**

**Problem**

A mysterious illness drives skin inflammation in fishermen and bathing people.

**Results**

Here we identified the marine algal toxin Portimine A produced by the dinoflagellate *Vulcanodinium rugosum* as a driver of skin inflammation in humans. Specifically, PortimineA-inactivated translation promotes ZAKα- and P38 kinase-driven NLRP1 inflammasome response, which leads to the release of large amounts of pro-inflammatory cytokines IL-1β, IL-1α, and IL-18. Furthermore, genetic targeting of ZAKα alleviates Portimine-induced inflammation in human organotypic skins. Finally, zebrafish models of PortimineA exposure support the pathological contribution of ZAKα/ NLRP1 pathway observed in humans

**Impact**

Our findings point to Portimine A produced by the dinoflagellate *Vulcanodinium rugosum* as the cause of the dermatitis observed in bathing people and fishermen in various geographical areas. In addition, unraveling the critical mechanism of action engaged allowed determining ZAKα kinase and NLRP1 as critical pharmacological targets that support PortimineA-driven skin inflammation.

Log-rank test for trend with Bonferroni correction were used. For caudal fin damage experiments (tail area), we used one-way ANOVA (Kruskal–Wallis test). *P* values are shown in figures and are linked to the following meaning *** when $P \leq 0.0001$. No blinding or randomization were performed.

## Data availability

Experimental dataset of toxin profile in sea samples are availabale in the following database: Plessis et al (2024) Data set.

The source data of this paper are collected in the following database record: biostudies:S-SCDT-10_1038-S44321-025-00197-4.

## Peer review information

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

## Acknowledgements

The authors acknowledge Captain Dieye Ndongo of the Senegalese Gendarmerie (2020) and Anthony Nehmé (2021) for granting access to facilities at sea. The authors also thank the National Senegalese small-scale fishermen committee "CLPA" of Thiaroye (2021) and Ndayane (2020–2021) for facilitating exchanges with fishermen. The Senegalese ministry of environement (Ministère de l'Environnement et du Développement Durable) via Lamine Diame for their early encouragment. Special thanks to Dr. Cheikh Sokhna and Hubert Bassène (IRD MINES) for providing access to laboratory facilities and for their early support during the crisis. The authors acknowledge all lab members for their critical advices and support on this project. The authors also acknowledge the IPBS microscopy platform and the UMU aquatic animal facility. The Imaging Core Facility TRI-IPBS received financial support from ITMO Cancer Aviesan (Alliance Nationale Pour les Sciences de la Vie et de la Santé, National Alliance for Life Science and Health) within the framework of the Cancer Plan. This project was supported by the Fondation pour la Recherche Médicale "Amorçage Jeunes Equipes" (AJE20151034460), the Agence Nationale de la Recherche (ANR-PSICOPAK and ANR-INFLAMATOX), and the European Research Council (StG INFLAME 804249) to E Meunier, the ANR-COMETH- to C Cougoule and P Hess, an Invivogen-Conventions industrielles de formation par la recherche PhD grant to L Ravon-Kattowsky, and the "la Caixa" Foundation (LCF/PR/HR21/52410027), MCIN (PID2022-139754OB-I00), and MCIN and "ESF Investing in your Future" Fellowship (RYC2019-571 027799-I) to FJ Roca. The authors acknowledge funding to L Plessis through Ifremer—Groupe Rocher convention and ANRT funding, as well as Ifremer core funding (PHYTOX-R306-01) to P Hess, D Réveillon, F Hervé and K Lhaute. This study benefitted from the contribution of the Awatox demonstration project (IRD,/BMBF SRFC; grant 01DG12073E) and the Art Sunu Gueej initiative. John Common and Stephen Wearne are supported by the Asian Skin Microbiome Programme 2.0 IAF-PP grant (H22J1a0040). John Common is supported by NIHR Newcastle Biomedical Research Centre. FLZ's laboratory is funded by the National Medical Research Council, Singapore (MOH-001499), Ministry of Education Tier 1 grant (RT23/23), and Ministry of Education Tier 2 grant (MOE-T2EP30222-0008). The funders had no role in study design, data collection and analysis, decision to publish, or preparation of the manuscript.

## Author contributions

**Léana Gorse**: Conceptualization; Data curation; Formal analysis; Validation; Investigation; Visualization; Methodology; Writing—original draft; Writing—review and editing. **Loïc Plessis**: Conceptualization; Data curation; Formal analysis; Validation; Investigation; Visualization; Methodology; Writing—original draft; Writing—review and editing. **Stephen Wearne**: Conceptualization; Data curation; Formal analysis; Validation; Investigation; Visualization; Methodology; Writing—original draft; Writing—review and editing. **Margaux Paradis**: Conceptualization; Data curation; Formal analysis; Validation; Investigation; Visualization; Methodology; Writing—original draft; Writing—review and editing. **Miriam Pinilla**: Conceptualization; Data curation; Formal analysis; Validation; Investigation; Visualization; Methodology; Writing—original draft; Writing—review and editing. **Rae Chua**: Data curation; Formal analysis; Validation; Investigation; Visualization; Methodology. **Seong Soo Lim**: Conceptualization; Data curation; Formal analysis; Validation; Investigation; Visualization; Methodology; Writing—original draft; Writing—review and editing. **Elena Pelluz**: Conceptualization; Data curation; Formal analysis; Validation; Investigation; Visualization; Methodology; Writing—original draft; Writing—review and editing. **Gee-Ann TOH**: Investigation; Methodology. **Raoul Mazars**: Investigation; Methodology. **Caio Bomfim**: Investigation; Methodology. **Fabienne Hervé**: Supervision; Investigation; Methodology; Writing—original draft. **Korian Lhaute**: Investigation; Methodology. **Damien Réveillon**: Supervision; Investigation; Methodology; Writing—original draft. **Bastien Suire**: Investigation; Methodology. **Léa Ravon-Katossky**: Investigation; Methodology. **Thomas Benoist**: Investigation; Methodology. **Léa Fromont**: Investigation; Methodology. **David Péricat**: Investigation; Methodology. **Kenneth Neil Mertens**: Investigation; Methodology. **Amélie Derrien**: Investigation; Methodology. **Aouregan Terre-Terrillon**: Investigation; Methodology. **Nicolas Chomérat**: Investigation; Methodology. **Gwenaël Bilien**: Investigation; Methodology. **Véronique Séchet**: Investigation; Methodology. **Liliane Carpentier**: Investigation; Methodology. **Mamadou Fall**: Investigation; Methodology. **Amidou Sonko**: Investigation; Methodology. **Hadi Hakim**: Investigation; Methodology. **Nfally Sadio**: Funding acquisition; Investigation; Methodology. **Jessie Bourdeaux**: Supervision; Validation; Investigation; Visualization; Methodology. **Céline Cougoule**: Funding acquisition; Validation; Investigation; Visualization; Methodology. **Anthony K Henras**: Conceptualization; Data curation; Formal analysis; Supervision; Funding acquisition; Validation; Investigation; Visualization; Methodology; Writing—original draft; Project administration; Writing—review and editing. **Ana Belen Perez-Oliva**: Conceptualization; Data curation; Formal analysis; Supervision; Funding acquisition; Validation; Investigation; Visualization; Methodology; Writing—original draft; Writing—review and editing. **Patrice Brehmer**: Data collection; Conceptualization; Data curation; Formal analysis; Supervision; Funding acquisition; Validation; Investigation; Visualization; Methodology; Writing—original draft; Project administration; Writing—review and editing. **Francisco J Roca**: Conceptualization; Data curation; Formal analysis; Supervision; Funding acquisition; Validation; Investigation; Visualization; Methodology; Writing—original draft; Writing—review and editing. **Franklin L Zhong**: Conceptualization; Data curation; Formal analysis; Supervision; Funding acquisition; Validation; Investigation; Visualization; Methodology; Writing—original draft; Project administration; Writing—review and editing. **John Common**: Conceptualization; Data curation; Formal analysis; Supervision; Funding acquisition; Validation; Investigation; Visualization; Methodology; Writing—original draft; Project administration; Writing—review and editing. **Etienne Meunier**: Conceptualization; Data curation; Formal analysis; Supervision; Funding acquisition; Validation; Investigation; Visualization; Methodology; Writing—original draft; Project administration; Writing—review and editing. **Philipp Hess**: Conceptualization; Data curation; Formal analysis; Supervision; Funding acquisition; Validation; Investigation; Visualization; Methodology; Writing—original draft; Project administration; Writing—review and editing.

Source data underlying figure panels in this paper may have individual authorship assigned. Where available, figure panel/source data authorship is listed in the following database record: biostudies:S-SCDT-10_1038-S44321-025-00197-4.

## Disclosure and competing interests statement

The authors declare no competing interests.

# Expanded View Figures

**Figure EV1.   (refers to Fig. 1):** *Vulcanodinium rugosum*-**produced Portimine A triggers human skin epithelial cell necrosis and IL-1 cytokine release.** ▶

(**A**) Amplification results of qPCR assay including derivative melting curves plot and amplification curves plot. The qPCR analysis was performed in duplicate. The melting temperature (Tm) was calculated to be 80.3 °C for the strain and sample and the cycle threshold was of 24 and 31 cycles, respectively. (**B**) Cytokine analysis 24 h after exposure of primary human keratinocytes (pHEKs) to purified extracts (Sample II isolated in Fig. 1C, dilution 1/20,000 from the isolated fraction). Representative experiment of three independent experiments. (**C**) Cytotoxicity of Pinnatoxin H and -G, Portimine A and -B pHEK. 24 h treatment, $n = 3$, mean ± SEM. (**D**) Cytotoxicity of extracts from *V. rugosum* cultures (IFR-VRU-01) and biomass sampled in Senegal (2020) on pHEK, expressed as PortimineA equivalent concentrations (determined by LC-MS/MS), 24 h treatment, $n = 3$, mean ± SEM. (**E**) Cell lysis (LDH) evaluation in pHEKs after 30 h exposure to Sample II, pure Portimine A (1,7 ng/mL, calculated from Sample II $IC_{50}$ in **D**), Portimine B (1,7 ng/mL, calculated from Sample II $IC_{50}$ in **D**), pure 400 ng/mL Pinnatoxin H and -G or combinations of all those toxins by always keeping a final concentration of 1.7 nM of Portimine A or Poritmine B in the different mixtures generated. ***$P \le 0.0001$, two-way ANOVA with multiple comparisons. Values are expressed as mean ± SEM. Graphs show one experiment performed in triplicates at least three times. (**F**) Plasma membrane permeabilization (SYTOX Green incorporation, 9 h) in pHEKs after exposure to Sample II (1/20,000), Portimine A (4 ng/mL), Portimine B (400 ng/mL) or Pinnatoxin G/H (40 ng/mL). ***$P \le 0.0001$, two-way ANOVA with multiple comparisons. Values are expressed as mean ± SEM. Graphs show one experiment performed in triplicates at least three times. (**G**) SYTOX Green incorporation in pHEKs, primary human endothelial, nasal, monocytes, lymphocytes or neutrophils cells 6 h after exposure to Portimine A (4 ng/mL) or Portimine B (4 ng/mL). ***$P \le 0.0001$, two-way ANOVA with multiple comparisons. Values are expressed as mean ± SEM. Graphs show one experiment performed in triplicates at least three times.

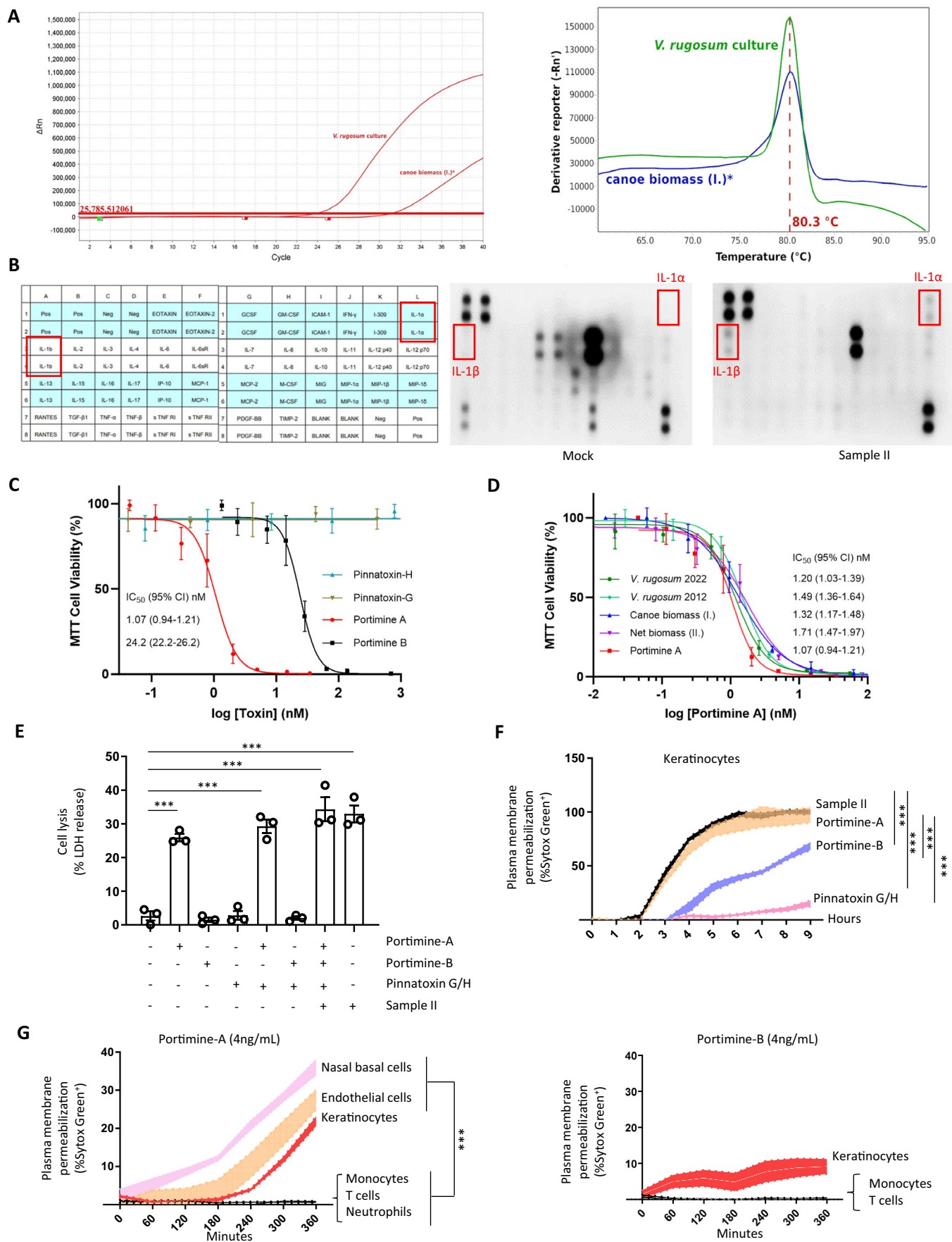

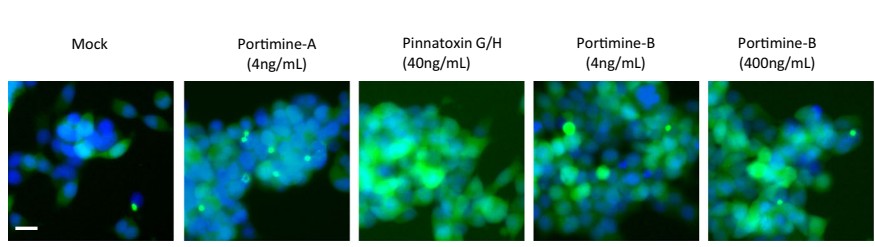

HEK 293 NLRP1/ASC-GFP

ASC-GFP Nuclei (Hoechst)

**Figure EV2.   (refers to Fig. 2): Portimine A activates the NLRP1 inflammasome in human skin epithelial cells.**

Fluorescence micrographs of ASC-GFP specks in HEK293T cells individually expressing NLRP1 and exposed to 4 ng/mL of Portimine A, 400 ng/mL of Pinnatoxin H/G or to 4 ng/mL or 400 ng/mL of Portimine B for 12 h. ASC-GFP (green) pictures were directly taken after adding Hoechst (nuclei staining). Images shown are from one experiment and are representative of $n = 3$ independent experiments. Scale bar, 10 μm.

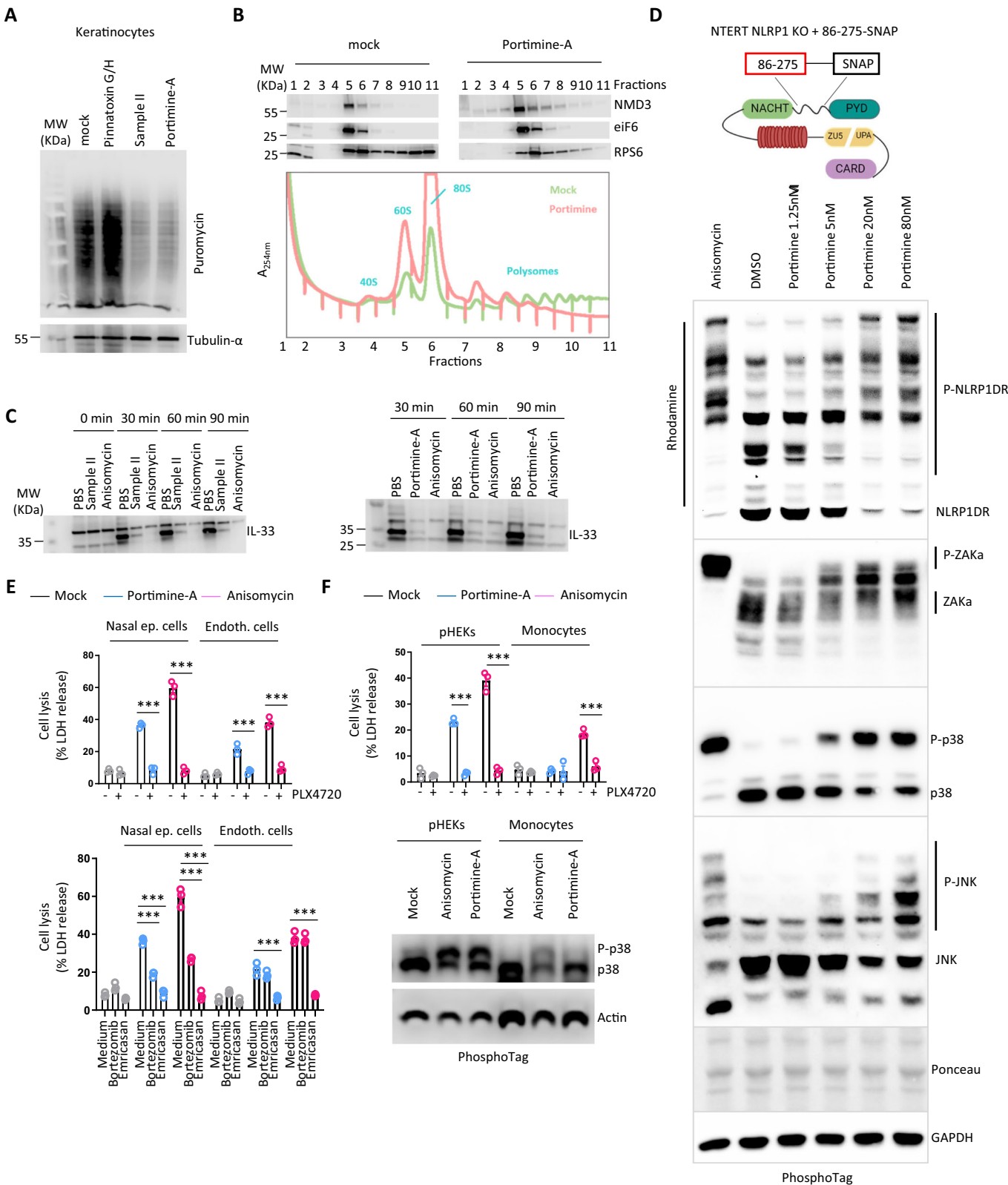

◀ **Figure EV3.   (refers to Fig. 3): Portimine-inhibited translation promotes ZAKα-dependent P38 activation and hNLRP1 inflammasome activation in epithelial cells.**

(A) Determination of protein synthesis in pHEKs in response to Sample II (1/20000 dilution), Portimine A (4 ng/mL) or Pinnatoxins-H/G (40 ng/mL) by measuring puromycin incorporation after 10 h exposure. Immunoblots show lysates from one experiment performed at least three times. (B) Ribosome profiling and ribosomal fraction analysis after exposing HEK293 cells expressing or not NLRP1 to Portimine A (4 ng/mL) for 2 h. Images and Immunoblotting are representatives of one experiment performed at least three times. (C) In vitro translation of the reporter plasmid coding for Interleukin 33 (IL-33) by rabbit reticulocyte lysates in the presence/absence of Sample II (1/20,000 dilution), Portimine A (4 ng/mL), Anisomycin (1 μg/mL). Immunoblotting are representatives of one experiment performed at least three times. (D) Phosphotag blotting of phosphorylated ZAKα, P38, JNK and NLRP1 disordered Region (DR) in NTERT NLRP1 KO + 86-275-SNAP cells exposed to various amounts of Portimine A or to the known RSR inducer Anisomycin (1 μg/mL) for one hour. Ponceau staining and GAPDH were used as internal protein loading controls. Immunoblots show lysates from one experiment performed at least two times. (E, F) Phosphotag blotting of phosphorylated P38 and cell lysis (LDH) evaluation in pHEKs, endothelial cells, nasal basal cells and human blood monocytes after 24 h exposure to pure Portimine A (4 ng/mL) or Anismoycin (1 μg/mL and 10 μg/mL in monocytes). When specified the compounds PLX4720 (ZAKα, 10 μM), Emricasan (pan Caspase inhibitor, 5 μM) and bortezomib (proteasome inhibitor, 1 μM) were used. ***$P \leq 0.0001$, two-way ANOVA with multiple comparisons. Values are expressed as mean ± SEM. Graphs show one experiment performed in triplicates at least three times.

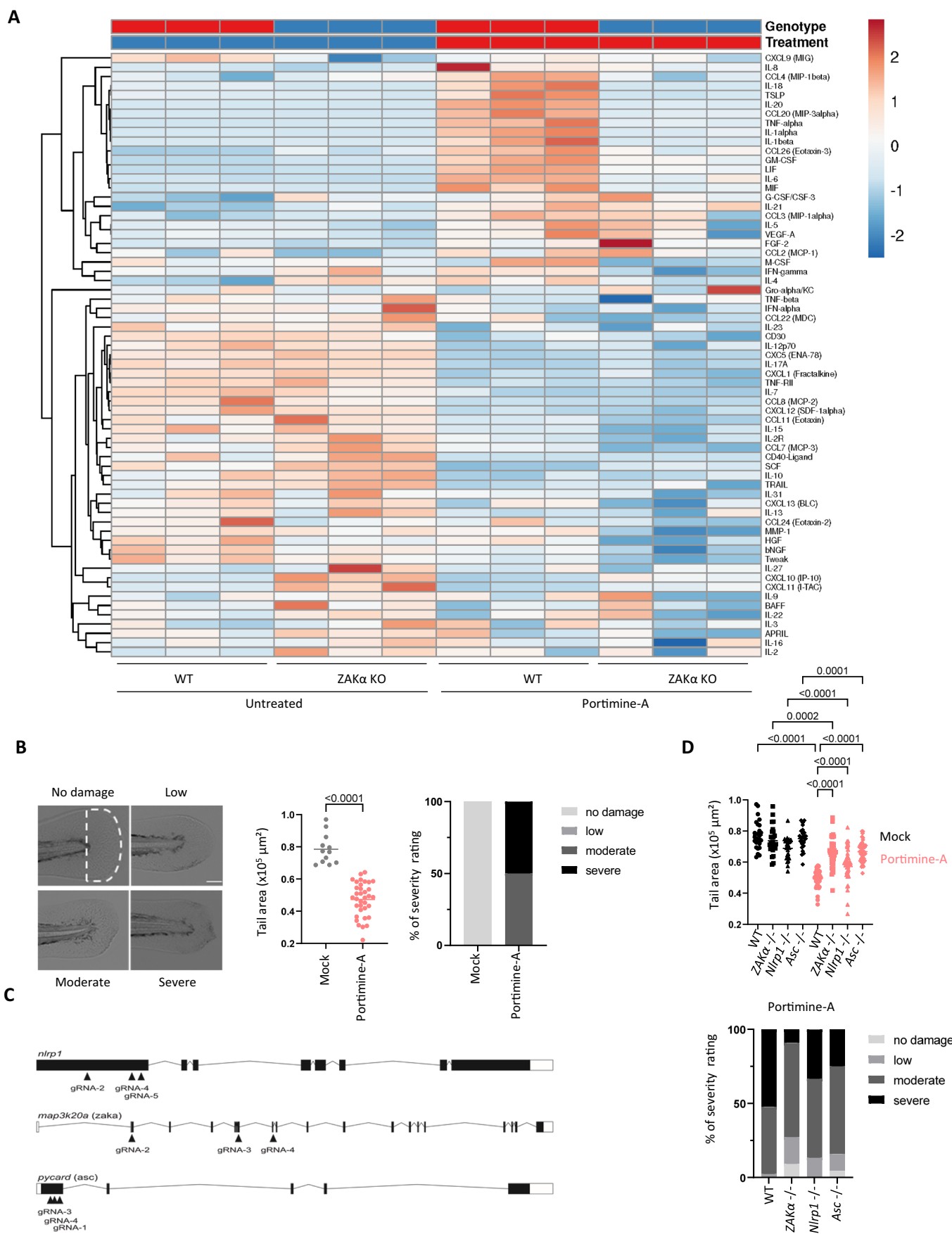

◄ **Figure EV4.  (refers to Fig. 4): ZAKα and NLRP1 contribution to Portimine-induced skin inflammation in 3D skin and zebrafish models.**

(A) 65 cytokine analyzed in 3D skin treated or not with Portimine A. (B) Determination and quantification of zebrafish larvae fin tail damage induced by Portimine A (75 nM) in WT larvae (20/group). Two specific parameters were studied for damages quantifications, namely the Tail area ($\times 10^5\ \mu m^2$) and the % of damage severity. Scale bar 100 μm. *P* values indicated in figure, one-way ANOVA (Kruskal–Wallis test). Graphs show one experiment performed three times. (C) CRISPR/Cas9 gRNA strategy used to genetically ablate ZAKα, NLRP1 and Asc in zebrafish embryo (for details see material and methods). (D) Determination and quantification of zebrafish larvae fin tail damage induced by Portimine A (75 nM) in WT, *Nlrp1-, Asc-* and *ZAKa*-deficient larvae (20/group) after 30 h. *P* values indicated in figure, one-way ANOVA (Kruskal–Wallis test). Scale bar 100 μm. Graphs show one experiment performed two times.

