## [Peer Review File · EMBO Molecular Medicine]

Portimine A toxin causes skin inflammation through ZAK α -dependent NLRP1 inflammasome activation

Leana Gorse, Loïc Plessis, Stephen Wearne, Margaux Paradis, Miriam Pinilla, Rae Chua, Seong Lim, Elena Pelluz, Gee Ann Toh, Raoul Mazars, Caio Barbosa Bomfim, Fabienne Hervé, Korian Lhaute, Damien Réveillon, Bastien Suire, Léa Ravon-Katosky, Thomas Benoist, Léa Fromont, David Pericat, Kenneth Mertens, Amélie Derrien, Aouregan Terre-Terrillon, Nicolas Chomérat, Gwenaël Bilien, Véronique Séchet, Liliane Carpentier, Mamadou Fall, Amidou Sonko, Hadi Hakim, Nfally Sadio, Jessie Bourdeaux, celine cougoule, Anthony Henras, Ana Perez-Oliva, Patrice Brehmer, Francisco Roca, Franklin Zhong, John Common, Etienne Meunier, and Philipp Hess

Corresponding authors: Etienne Meunier (etienne.meunier@ipbs.fr), Franklin Zhong (franklin.zhong@ntu.edu.sg), Patrice Brehmer (patrice.brehmer@ird.fr), Francisco Roca (fjroca@um.es), John Common (john.common@newcastle.ac.uk), Philipp Hess (philipp.hess@ifremer.fr)

Review Timeline:

Submission Date:	31st Jul 24
Editorial Decision:	3rd Sep 24
Revision Received:	10th Dec 24
Editorial Decision:	15th Jan 25
Revision Received:	21st Jan 25
Authors' Correspondence:	23rd Jan 25
Accepted:	23rd Jan 25

Editor: Lise Roth

Transaction Report:

3rd Sep 2024

Dear Dr. Meunier,

Thank you for the submission of your manuscript to EMBO Molecular Medicine, and please accept my apologies for the delay in getting back to you in this busy time of the year. We have now received feedback from the three reviewers who agreed to evaluate your manuscript. As you will see from the reports below, the referees acknowledge the interest of the study and are overall supporting publication of your work pending appropriate revisions.

Addressing the reviewers' concerns in full will be necessary for further considering the manuscript in our journal, and acceptance of the manuscript will entail a second round of review. EMBO Molecular Medicine encourages a single round of revision only and therefore, acceptance or rejection of the manuscript will depend on the completeness of your responses included in the next, final version of the manuscript. For this reason, and to save you from any frustrations in the end, I would strongly advise against returning an incomplete revision.

We are expecting your revised manuscript within three months, if you anticipate any delay, please contact us.

We require:

4) A .docx formatted letter INCLUDING the reviewers' reports and your detailed point-by-point responses to their comments. As part of the EMBO Press transparent editorial process, the point-by-point response is part of the Review Process File (RPF), which will be published alongside your paper.

5) A complete author checklist, which you can download from our author guidelines (<https://www.embopress.org/page/journal/17574684/authorguide#submissionofrevisions>). Please insert information in the checklist that is also reflected in the manuscript. The completed author checklist will also be part of the RPF.

6) All Materials and Methods need to be described in the main text using our 'Structured Methods' format, which is required for all research articles. According to this format, the Methods section includes a Reagents and Tools Table (listing key reagents, experimental models, software and relevant equipment and including their sources and relevant identifiers) followed by a Methods and Protocols section describing the methods using a step-by-step protocol format. The aim is to facilitate adoption of the methodologies across labs. More information on how to adhere to this format as well as a downloadable template (.docx) for the Reagents and Tools Table can be found in our author guidelines:

<https://www.embopress.org/page/journal/17574684/authorguide#structuredmethods>

7) Please note that all corresponding authors are required to supply an ORCID ID for their name upon submission of a revised manuscript.

8) It is mandatory to include a 'Data Availability' section after the Materials and Methods. Before submitting your revision, primary datasets produced in this study need to be deposited in an appropriate public database, and the accession numbers and database listed under 'Data Availability'. Please remember to provide a reviewer password if the datasets are not yet public (see <https://www.embopress.org/page/journal/17574684/authorguide#dataavailability>).

9) For data quantification: please specify the name of the statistical test used to generate error bars and P values, the number (n) of independent experiments (specify technical or biological replicates) underlying each data point and the test used to calculate p-values in each figure legend. The figure legends should contain a basic description of n, P and the test applied. Graphs must include a description of the bars and the error bars (s.d., s.e.m.). Please provide exact p values.

10) Our journal encourages inclusion of *data citations in the reference list* to directly cite datasets that were re-used and obtained from public databases. Data citations in the article text are distinct from normal bibliographical citations and should directly link to the database records from which the data can be accessed. In the main text, data citations are formatted as follows: "Data ref: Smith et al, 2001" or "Data ref: NCBI Sequence Read Archive PRJNA342805, 2017". In the Reference list, data citations must be labeled with "[DATASET]". A data reference must provide the database name, accession number/identifiers and a resolvable link to the landing page from which the data can be accessed at the end of the reference. Further instructions are available at .

11) We replaced Supplementary Information with Expanded View (EV) Figures and Tables that are collapsible/expandable online. A maximum of 5 EV Figures can be typeset. EV Figures should be cited as 'Figure EV1, Figure EV2' etc... in the text and their respective legends should be included in the main text after the legends of regular figures.

12) The paper explained: EMBO Molecular Medicine articles are accompanied by a summary of the articles to emphasize the major findings in the paper and their medical implications for the non-specialist reader. Please provide a draft summary of your article highlighting

13) Author contributions: CRedit has replaced the traditional author contributions section because it offers a systematic machine readable author contributions format that allows for more effective research assessment. Please remove the Authors Contributions from the manuscript and use the free text boxes beneath each contributing author's name in our system to add specific details on the author's contribution. More information is available in our guide to authors.

Please also suggest a visual abstract to illustrate your article as a PNG file 550 px wide x 300-600 px high.

16) As part of the EMBO Publications transparent editorial process initiative (see our Editorial at <http://embomolmed.embopress.org/content/2/9/329>), EMBO Molecular Medicine will publish online a Review Process File (RPF) to accompany accepted manuscripts.

In the event of acceptance, this file will be published in conjunction with your paper and will include the anonymous referee reports, your point-by-point response and all pertinent correspondence relating to the manuscript. Let us know whether you agree with the publication of the RPF and as here, if you want to remove or not any figures from it prior to publication. Please note that the Authors checklist will be published at the end of the RPF.

I look forward to receiving your revised manuscript.

Yours sincerely,

Lise Roth

***** Reviewer's comments *****

Referee #1 (Comments on Novelty/Model System for Author):

This study from Gorese, Plessis and colleagues focuses on understanding the etiology of an inflammatory skin disease that afflicted a group of Senegalese Fisherman in 2020-2021. Using standard genomic sequencing approaches, the authors identified a marine dinoflagellate *Vulcanodinium rugosum* in environmental samples from sites where this illness was observed. They further demonstrate that a toxin (Portimine A) produced by this dinoflagellate induces an inflammatory cell death process known as pyroptosis in human epithelial keratinocytes resulting in inflammation and epithelial damage in human skin organoid cultures. Using genetic and biochemical approaches, they convincingly demonstrate that Portimine A is a new activator of the ribotoxic stress induced ZAK α dependent pathway that engages the NLRP1 inflammasome in human epithelial cells. Further, they provide evidence that Portimine A can induce epithelial damage and death in zebrafish via a ZAK α and NLRP1-dependent process providing important in vivo observations. Overall, the study is laid out in a logical manner, is well controlled, and was very interesting to read. I have a few minor comments (listed below) that should be addressed.

1. Line 152: The authors should think about changing "Ninjurin-1-driven cell shrinkage" to "Ninjurin-1-driven cell lysis" to more accurately describe NINJ1 activity.
2. Figure 2B: It would be helpful to include Fraction numbers below the polysome profile to help readers interpret the immunoblots provided above the image.
3. Figure 2E: labels indicating the genetic background of each lane appear to be missing from the panel.
4. The authors conclude from the immunoblots provided in Figure 2G that the disordered (DR) region of NLRP1 becomes phosphorylated following Portimine A treatment, but the provided blots are not very convincing. Inclusion of a positive control for DR phosphorylation, such as anismoycin or UVB, may help to convince that phosphorylation is observed in Portimine A-treated cells.
5. In Figure 2I/J the authors provide evidence that S107F and P108-T112 are critical amino acids for Portimine A induced inflammasome activation (ASC specks). To confirm specificity for the ribotoxic stress response, the authors should demonstrate that these mutations do not affect VbP induced ASC speck formation.
6. The authors should increase the size of Figure EV3A so that the protein names can be read more easily.

Referee #2 (Comments on Novelty/Model System for Author):

rodents are not suitable for the study of NLRP1 activation. The pathway has evolved toward a different repertoire of NLRP1 genes and functions in rodents

Referee #2 (Remarks for Author):

The study by Gorse et al. presents significant findings on how the marine dinoflagellate *Vulcanodinium rugosum* causes acute dermatitis. The key discovery revolves around the toxin Portimine A, produced by this dinoflagellate, which impairs translation in human keratinocytes. This impairment triggers the activation of stress kinases, specifically ZAK α , which in turn activates the NLRP1 inflammasome, leading to inflammation. The study's use of human skin models and zebrafish further validates that Portimine A-mediated inflammation is dependent on NLRP1.

This research contributes meaningfully to the growing body of evidence that translation defects can activate the NLRP1

inflammasome, promoting inflammation. The findings are particularly relevant as they provide another example of how environmental toxins can affect human skin health at the molecular level.

Technical Comments:

Selective Activation of ZAK α and NLRP1: The authors have demonstrated that Portimine A induces cell death in endothelial and epithelial cells, but not in monocytes or other immune cells. To strengthen their hypothesis that only these cell types can respond to the toxin via ZAK α and NLRP1, it would be crucial to show that these pathways are not activated in monocytes or other immune cells. This could be achieved through additional experiments comparing ZAK α and NLRP1 activation across various cell types.

ASC Speck Formation in Physiologically Relevant Models: The use of 293T cells expressing ASC-GFP constructs to monitor ASC speck formation is a valid approach. However, given the physiological relevance, it would be more compelling if the authors could demonstrate ASC speck formation in their primary human keratinocytes (pHEKs) model. Additionally, comparing this with their NLRP1 knockout models would provide a robust confirmation of their findings.

Investigation of Portimine B: The study detected both Portimine A and B in environmental samples, and the authors noted that Portimine B also induces ZAK α phosphorylation. However, they did not assess whether Portimine B can induce ZAK α and NLRP1-mediated inflammation in the skin models or zebrafish. Including such experiments would be beneficial to determine if both toxins share similar mechanisms of action. Moreover, exploring the effects of Portimine B on immune cells, such as monocytes and neutrophils, would be of interest, in case it differs from Portimine A.

Phostag and Controls: In Figure 2G, the visibility of phosphorylation is weak. It would be helpful to include anisomycin as a positive control for phosphorylation of the disordered region. This control could also be used in the ASC speck quantification below. Additionally, the use of phosphorylation mutants as a negative control would enhance the reliability of the phosphorylation data presented.

Minor Comments:

Figure 1B: The legend for Figure 1B should be clearly indicated on the panel. This would improve the clarity and understanding of the figure for readers.

Figure 2E: In Figure 2E, the genotype (ZAK α deficiency/proficiency) is not shown on the panel. Adding this information is necessary to understand the experimental conditions.

Line 244: The reference in line 244 is shown twice.

Referee #3 (Remarks for Author):

Gorse et al attempt to elucidate the mechanism of an unexplained human skin disease in Senegalese fisherman. They begin the study by finding the natural product protamine A in certain marine samples, and then investigate if this agent could cause of the human skin disease. Using a variety of models, they show that Portimine A can activate the human NLRP1 inflammasome and cause inflammatory cell death in human primary keratinocytes. They convincingly show that portimine activates the ZAK α pathway to trigger human NLRP1 assembly (a pathway that is already known). However, the data that Portimine activates the NLRP1 inflammasome in zebrafish is weak, as only the slightest change in pathology is observed. Moreover, there is no direct connection that the actual skin disease is caused by Portimine itself.

Overall, the report is well written and the conclusions that portimine can activate NLRP1 in experimental models are convincing. However, this work does not show that protamine is a general activator of NLRP1 in fish, nor does it definitely show that portimine actually causes a human skin disease.

Major comments:

1. The authors claim that portimine A induces cell death in endothelial and nasal epithelial cells, but not in blood-isolated monocytes, neutrophils and lymphocytes. As portimine clearly has non-NLRP1 dependent toxicity, it is important to show that this death is due NLRP1 (confirming NLRP1 is present and activity using Western Blot and Val-boroPro, that it is blocked by VX and proteasome inhibitors, etc...)

2. The authors clearly show that Portimine A, like anisomycin, induces some non-NLRP1 dependent cell death (Fig. 1H). As such, they should include a blot for cleaved N-terminal GSDMD and a marker of apoptosis (PARP, cleaved casp-3) in these experiments.

3. They identify several NLRP1 mutations that confer resistance to portimine. They should confirm that NLRP1 function is still

intact by showing responsiveness to Val-boroPro in these lines.

4. The authors should perform genome mining for marine species for NLRP1 (or related) proteins that contain the ZAKa motif, so that they can speculate on the species (in marine environments) that may respond to Portimine A. Especially as this motif is not conserved in mice.

5. The data that NLRP1 is responsible for Portimine pathology in zebrafish is weak (only small changes seen), and it is not clear how statistically significant this is. Far more data, including evidence of pyroptosis (GSDMD cleavage), is needed to confirm that this natural product really activates the NLRP1 inflammasome in these fish.

6. While the authors show that portimine a is in the marine samples, and that portimine a can activate human NLRP1, they do not definitively show that protamine A is indeed responsible for the skin disease. The discussion should make clear that this was not unequivocally shown.

Other:

Fig3A, typo: "liquide"

***** Reviewer's comments *****

We thank our reviewers for their very helpful remarks and suggestions. Overall, they strongly improved our study.

Referee #1 (Comments on Novelty/Model System for Author):

This study from Gorese, Plessis and colleagues focuses on understanding the etiology of an inflammatory skin disease that afflicted a group of Senegalese Fisherman in 2020-2021. Using standard genomic sequencing approaches, the authors identified a marine dinoflagellate *Vulcanodinium rugosum* in environmental samples from sites where this illness was observed. They further demonstrate that a toxin (Portimine A) produced by this dinoflagellate induces an inflammatory cell death process known as pyroptosis in human epithelial keratinocytes resulting in inflammation and epithelial damage in human skin organoid cultures. Using genetic and biochemical approaches, they convincingly demonstrate that Portimine A is a new activator of the ribotoxic stress induced ZAKa dependent pathway that engages the NLRP1 inflammasome in human epithelial cells. Further, they provide evidence that Portimine A can induce epithelial damage and death in zebrafish via a ZAKa and NLRP1-dependent process providing important in vivo observations. Overall, the study is laid out in a logical manner, is well controlled, and was very interesting to read. I have a few minor comments (listed below) that should be addressed.

We thank our reviewer for his/her analysis and comments.

1. Line 152: The authors should think about changing "Ninjurin-1-driven cell shrinkage" to "Ninjurin-1-driven cell lysis" to more accurately describe NINJ1 activity.

According to our reviewer suggestions, we directly addressed this point in the revised version of the manuscript, lines 148-149.

2. Figure 2B: It would be helpful to include Fraction numbers below the polysome profile to help readers interpret the immunoblots provided above the image.

According to our reviewer suggestions, we addressed this point in the revised version of the figures, now labelled as "Fig. EV3B".

3. Figure 2E: labels indicating the genetic background of each lane appear to be missing from the panel. According to our reviewer suggestions, we addressed this point in the revised version of the figures, now labelled as "Fig. 3D".

4. The authors conclude from the immunoblots provided in Figure 2G that the disordered (DR) region of NLRP1 becomes phosphorylated following Portimine A treatment, but the provided blots are not very convincing. Inclusion of a positive control for DR phosphorylation, such as anismoycin or UVB, may help to convince that phosphorylation is observed in Portimine A-treated cells.

We fully agree with our reviewer comment and accordingly performed new experiments by using a new version of the NLRP1 DR tag in immortalized keratinocytes (N/TERT) deficient for NLRP1 and complemented with NLRP1 disordered construct (referred to as "NTERT NLRP1 KO + 86-275-SNAP"). The methodology is explained in the figures below that has also been included in the new version of manuscript and are referenced to as "Fig3" and "Fig. EV3D". As shown in those figures, exposure of various amount of Portimine A and to Anisomycin (positive control of NLRP1 phosphorylation), we could observe through phosphotag analysis that Portimine A induces phosphorylation of the NLRP1 DR in a concentration-dependent manner as well as in a ZAKalpha (inhibitor of ZAKalpha used 6p, 1μM)-dependent manner. Note that P38 and JNK phosphorylation induced by portimine (and Anisomycin) are entirely dependent of ZAK activation as shown in presence of the 6p compound. Overall, we are confident that this new batch of results support the ability of Portimine A-induced ZAKalpha/P38 pathway to promote NLRP1 phosphorylation. These results are now included in the

new version of the manuscript in the **Fig 3** and **EV3** panels.

Figure: Phosphotag blotting of phosphorylated ZAKα, P38, JNK and NLRP1 disordered Region (DR) in NTERT NLRP1 KO + 86-275-SNAP cells exposed to various amounts of Portimine A or to the known RSR inducer Anisomycin (1μg/mL) for one hour. Ponceau staining and GAPDH were used as internal protein loading controls. Immunoblots show lysates from one experiment performed at least two times.

5. In Figure 2I/J the authors provide evidence that S107F and P108-T112 are critical amino acids for Portimine A induced inflammasome activation (ASC specks). To confirm specificity for the ribotoxic stress response, the authors should demonstrate that these mutations do not affect VbP induced ASC speck formation.

In agreement with our reviewer comment, we included our VbP controls in our figure. As shown below, they show that Vbp (10 μ M, 12 hours) still hold the capacity to promote NLRP1 response, independently of the variants studied, contrary to Portimine A. This strongly suggests that those variants are affecting the pathway of phosphorylation-driven NLRP1 activation but not the DPP8/9 regulatory pathway inhibited by VbP. The figure is now updated in the revised version of the study and referenced in the **Fig. 3** panel.

Figure: Fluorescence micrographs and respective quantifications of ASC-GFP specks in HEK293ASC-GFP reporter cells reconstituted with hNLRP1 or hNLRP1 plasmid constructs mutated for S107F and DelP108_T112 after 6 hours exposure to Portimine A (4ng/mL) or after 10 hours exposure to Val-boro-Pro (VbP, 10 μ M). ASC-GFP (green) pictures were taken in the dish after toxin exposure. Images shown are from one experiment and are representative of three independent experiments; scale bars, 10 μ m. ASC complex percentage was performed by determining the ratios of cells positive for ASC speckles (green, GFP) on the total nuclei (Hoechst). At least 10 fields from three independent experiments were analyzed. Values are expressed as mean \pm SEM. ***P \leq 0.001, two-way ANOVA with multiple comparisons. Graphs show one experiment performed in triplicate at least three times.

6. The authors should increase the size of Figure EV3A so that the protein names can be read more easily.

According to our reviewer suggestions, we addressed this point in the revised version of the figures

Referee #2 (Comments on Novelty/Model System for Author):

rodents are not suitable for the study of NLRP1 activation. The pathway has evolved toward a different repertoire of NLRP1 genes and functions in rodents

Referee #2 (Remarks for Author):

The study by Gorse et al. presents significant findings on how the marine dinoflagellate *Vulcanodinium rugosum* causes acute dermatitis. The key discovery revolves around the toxin Portimine A, produced by this dinoflagellate, which impairs translation in human keratinocytes. This impairment triggers the activation of stress kinases, specifically ZAK α , which in turn activates the NLRP1 inflammasome, leading to inflammation. The study's use of human skin models and zebrafish further validates that Portimine A-mediated inflammation is dependent on NLRP1.

This research contributes meaningfully to the growing body of evidence that translation defects can activate the NLRP1 inflammasome, promoting inflammation. The findings are particularly relevant as they provide another example of how environmental toxins can affect human skin health at the molecular level.

We thank our reviewer for his/her encouraging analysis and comments.

Technical Comments:

Selective Activation of ZAK α and NLRP1: The authors have demonstrated that Portimine A induces cell death in endothelial and epithelial cells, but not in monocytes or other immune cells. To strengthen their hypothesis that only these cell types can respond to the toxin via ZAK α and NLRP1, it would be crucial to show that these pathways are not activated in monocytes or other immune cells. This could be achieved through additional experiments comparing ZAK α and NLRP1 activation across various cell types.

As suggested by our reviewer, we performed new batch of experiments in primary human nasal basal cells, endothelial cells, monocytes and keratinocytes in presence or absence of Portimine A or with Anisomycin, a well know ribotoxic stress inducer. As shown in figures below, Anisomycin can trigger significant monocyte death but not Portimine A. In addition, PLX4720, an inhibitor of ZAK activity, strongly impairs Anisomycin-induced monocyte death, suggesting that the ZAKalpha pathway is well effective in monocytes and is unlikely to explain the lack of monocyte death in response to Portimine A. At this step, we do not have a clear idea about why such difference exists, but this has also been reported by the Baran group in a previous article (Tang *et al*, 2023). Accordingly, our groups are currently performing structural studies on the critical molecular targets of Portimine A which we expect will shed some light about the difference of responses between monocytes (and T cells, neutrophils) with keratinocytes, nasal epithelial cells and endothelial cells to Portimine A. Those results are now included in the **EV3 panel** of the revised manuscript.

Figure: A, B. Phosphotag blotting of phosphorylated P38 and cell lysis (LDH) evaluation in pHEKs, endothelial cells, nasal basal cells and human blood monocytes after 24 hours exposure to pure Portimine A (4ng/mL) or Anismoycin (1 μ g/mL and 10 μ g/mL in monocytes). When specified the compounds PLX4720 (ZAK α , 10 μ M), Emericasan (pan Caspase inhibitor, 5 μ M) and bortezomib (proteasome inhibitor, 1 μ M) were used. ***P \leq 0.001, two-way ANOVA with multiple comparisons. Values are expressed as mean \pm SEM. Graphs show one experiment performed in triplicates at least three times.

ASC Speck Formation in Physiologically Relevant Models: The use of 293T cells expressing ASC-GFP constructs to monitor ASC speck formation is a valid approach. However, given the physiological relevance, it would be more compelling if the authors could demonstrate ASC speck formation in their primary human keratinocytes (pHEKs) model. Additionally, comparing this with their NLRP1 knockout models would provide a robust confirmation of their findings.

Figure: Florescence micrographs and associated quantifications of ASC specks in pHEKs and NLRP1-deficient pHEKs generated in B and exposed or not to Portimine A (4ng/mL) for 24 hours. Hoechst (nuclei staining), ASC (anti-ASC antibody, green). Images shown are from one experiment and are representative of n = 3 independent experiments. Scale bar, 10 μ m. The percentage of ASC complex was performed by determining the ratios between cells positive for ASC speckles and the total of cell nuclei (Hoechst). At least 10 fields from each experiment were analyzed. Values are expressed as mean \pm SEM. ***P \leq 0.001, one-way ANOVA.

Investigation of Portimine B: The study detected both Portimine A and B in environmental samples, and the authors noted that Portimine B also induces ZAK α phosphorylation. However, they did not assess whether Portimine B can induce ZAK α and NLRP1-mediated inflammation in the skin models or zebrafish. Including such experiments would be beneficial to determine if both toxins share similar mechanisms of action. Moreover, exploring the effects of Portimine B on immune cells, such as monocytes and neutrophils, would be of interest, in case it differs from Portimine A.

Indeed, this is an interesting and important aspect raised by our reviewer. As determined in our biological samples, experiments and by a previous study from the Baran group (Tang *et al*, 2023), Portimine B is found in fewer amounts compared to Portimine A (Fig. A) in biological samples and Portimine B exhibits very low capacity at promoting cell death at the same concentrations than Portimine A (Figures below B-G). For instance, we had to add 100 time more Portimine B than Portimine A to see a ZAK α activation in our cell types. This strongly suggest to us that Portimine B, given its very low potency and its very low amounts compared to Portimine A, is unlikely to be responsible of the dermatitis observed in the fishermen. According, when we reconstituted the samples according to the cell death IC₅₀ calculated from Sample II, in figure B, and determined their ability to induce cell death on keratinocytes, only Portimine A-containing samples was able to trigger cell death (with a concentration of 1.7 nM). All these results have been included in the revised manuscript in Fig. EV1 and EV2.

Those results are discussed as follow in the revised version of the manuscript

“Furthermore, pure toxins PnTX-G and -H, Portimine A and B, as well as extracts from cultured *V. rugosum* and Senegalese biomass (samples I. and II.) were tested for pHEK cytotoxicity (MTT-assay). Portimine A exhibited a ca. 20-times lower IC₅₀ compared to Portimine B, i.e. significantly greater toxicity with an IC₅₀ of around 1 nM (Fig. EV1C, D). By expressing the extract concentration in Portimine A equivalents (determined by LC-MS/MS), their dose-response curves were very similar to that of Portimine A, obtaining IC₅₀-values in a comparable range, i.e. between 1.07 and 1.71 nM (Fig. EV1D). These results strongly suggested that the cytotoxic effects of the extracts may be primarily driven by the presence of Portimine A, potentially masking the weaker activity of Portimine B, which is present at significantly lower concentrations (N.B. Portimine B is found 27- to 50-fold less concentrated than Portimine A in environmental samples (Fig. 1C)). Relying on the IC₅₀ observed with Sample II (between 1.07 and 1.71 nM) (Fig. EV1D), we generated mixtures of Pinnatoxins G/H containing 1.7nM of Portimine A or of 1.7nM of Portimine B or with Pinnatoxins G/H + 1.7nM Portimine A + 1.7nM Portimine B and analyzed cell lysis (LDH release) in pHEKs (Fig. EV1E). The results showed that only the mixtures with Portimine A induced cell lysis in conditions mimicking the concentration of each toxin analyzed in the Sample II, which suggests that Portimine B exhibits very low activity compared to Portimine A for yet to be determined reason (Fig. EV1E). This is in agreement with recent findings from the Baran group that also determined a very low toxic potency of chemically synthesized Portimine B on various cell lines (Tang *et al*, 2023) and our own observations that Portimine B needs to be strongly concentrated to induce detectable keratinocyte death (Fig. EV1C-F). Thus, both the low concentration of Portimine B in the environmental samples and its extremely low potency at triggering cell death pointed to Portimine A as a putative causative agent responsible of skin inflammation and damages observed in fishermen.”

Figure: **A.** LC-MS/MS chromatogram of toxin profile (sample I., 2020) and chemical structures of Portimine A and Pinnatoxin H and concentrations of *V. rugosum* toxins in the samples I. to V., quantified by LC-MS/MS. All data from 2021 are indicated with an asterisk*. PnTX: Pinnatoxin, Port: Portimine, Iso: isomer. **B.** Cytotoxicity of Pinnatoxin H and -G, Portimine A and -B pHEK. 24h treatment, n=3, mean \pm SEM. **C.** Cytotoxicity of extracts from *V. rugosum* cultures (IFR-VRU-01) and biomass sampled in Senegal (2020) on pHEK, expressed as Portimine A equivalent concentrations (determined by LC-MS/MS), 24h treatment, n=3, mean \pm SEM. **D.** Cell lysis (LDH) evaluation in pHEKs after 30 hours exposure to Sample II, pure Portimine A (1,7ng/mL, calculated from Sample II IC50 in D), Portimine B (1,7ng/mL, calculated from Sample II IC50 in D), pure 400ng/mL Pinnatoxin H and -G

or combinations of all those toxins by always keeping a final concentration of 1.7 nM of Portimine A or Portimine B in the different mixtures generated. *** $P \leq 0.001$, two-way ANOVA with multiple comparisons. Values are expressed as mean \pm SEM. Graphs show one experiment performed in triplicates at least three times. **E.** Plasma membrane permeabilization (SYTOX Green incorporation, 9 hours) in pHEKs after exposure to Sample II (1/20000), Portimine A (4ng/mL), Portimine B (400ng/mL) or Pinnatoxin-G/H (40ng/mL). *** $P \leq 0.001$, two-way ANOVA with multiple comparisons. Values are expressed as mean \pm SEM. Graphs show one experiment performed in triplicates at least three times. **F.** SYTOX Green incorporation in pHEKs, primary human endothelial, nasal, monocytes, lymphocytes or neutrophils cells 6 hours after exposure to Portimine A (4ng/mL) or Portimine B (4ng/mL). *** $P \leq 0.001$, two-way ANOVA with multiple comparisons. Values are expressed as mean \pm SEM. Graphs show one experiment performed in triplicates at least three times. **G.** Florescence micrographs of ASC-GFP specks in HEK293T cells individually expressing NLRP1 and exposed to 4ng/mL of Portimine A, 400ng/mL of Pinnatoxin H/G or to 4ng/mL or 400ng/mL of Portimine B for 12 hours. ASC-GFP (green) pictures were directly taken after adding Hoechst (nuclei staining). Images shown are from one experiment and are representative of $n = 3$ independent experiments. Scale bar, 10 μm .

Phostag and Controls: In Figure 2G, the visibility of phosphorylation is weak. It would be helpful to include anisomycin as a positive control for phosphorylation of the disordered region. This control could also be used in the ASC speck quantification below. Additionally, the use of phosphorylation mutants as a negative control would enhance the reliability of the phosphorylation data presented.

We fully agree with our reviewer comment and accordingly performed new experiments by using a new version of the NLRP1 DR tag in immortalized keratinocytes (N/TERT) deficient for NLRP1 and complemented with NLRP1 disordered construct (referred to as "NTERT NLRP1 KO + 86-275-SNAP"). The methodology is explained in the figures below that have also been included in the new version of manuscript and are referenced to as "Fig3" and "Fig. EV3D". As shown in those figures, exposure of various amount of Portimine A and to Anisomycin (positive control of NLRP1 phosphorylation), we could observe through phosphotag analysis that Portimine A induces phosphorylation of the NLRP1 DR in a concentration-dependent manner as well as in a ZAKalpha (inhibitor of ZAKalpha used 6p, 1 μM)-dependent manner. Note that P38 and JNK phosphorylation induced by portimine (and Anisomycin) are entirely dependent of ZAK activation as shown in presence of the 6p compound. Overall, we are confident that this new batch of results support the ability of Portimine A-induced ZAKalpha/P38 pathway to promote NLRP1 phosphorylation. These results are now included in the new version of the manuscript in the **Fig 3** and **EV3** panels.

Figure: Phosphotag blotting of phosphorylated ZAK α , P38, JNK and NLRP1 disordered Region (DR) in NTERT NLRP1 KO + 86-275-SNAP cells exposed to various amounts of Portimine A or to the known RSR inducer Anisomycin (1 μ g/mL) for one hour. Ponceau staining and GAPDH were used as internal protein loading controls. Immunoblots show lysates from one experiment performed at least two times.

Minor Comments:

Figure 1B: The legend for Figure 1B should be clearly indicated on the panel. This would improve the clarity and understanding of the figure for readers.

According to our reviewer suggestions, we addressed this point in the revised version of the figure 1B.

Figure 2E: In Figure 2E, the genotype (ZAK α deficiency/proficiency) is not shown on the panel. Adding this information is necessary to understand the experimental conditions. According to our reviewer suggestions, we addressed this point in the revised version of the figures

Line 244: The reference in line 244 is shown twice. The reference has been removed in the revised manuscript

Referee #3 (Remarks for Author):

Gorse et al attempt to elucidate the mechanism of an unexplained human skin disease in Senegalese fisherman. They begin the study by finding the natural product protamine A in certain marine samples, and then investigate if this agent could cause of the human skin disease. Using a variety of models, they show that Portimine A can activate the human NLRP1 inflammasome and cause inflammatory cell death in human primary keratinocytes. They convincingly show that portimine activates the ZAKa pathway to trigger human NLRP1 assembly (a pathway that is already known). However, the data that Portimine activates the NLRP1 inflammasome in zebrafish is weak, as only the slightest change in pathology in observed. Moreover, there is no direct connection that the actual skin disease is caused by Portimine itself.

Overall, the report is well written and the conclusions that portimine can activate NLRP1 in experimental models are convincing. However, this work does not show that protamine is a general activator of NLRP1 in fish, nor does it definitely show that portimine actually causes a human skin disease.

We thank our reviewer for his/her analysis and insightful comments. We tried answering them in the rebuttal section blow.

Major comments:

1. The authors claim that portimine A induces cell death in endothelial and nasal epithelial cells, but not in blood-isolated monocytes, neutrophils and lymphocytes. As portimine clearly has non-NLRP1 dependent toxicity, it is important to show that this death is due NLRP1 (confirming NLRP1 is present and activity using Western Blot and Val-boroPro, that it is blocked by VX and proteasome inhibitors, etc...)

We acknowledge our reviewer comment.

- In this context, we performed additional experiments on endothelial, nasal basal cells and monocytes. As described in the figure below, nasal and endothelial cells exposed to Portimine A (and Anisomycin, control) undergo cell death in a ZAKalpha (PLX4720)- and Caspase (Emricasan)-dependent manner. However, they exhibit differential sensitivity to bortezomib (proteasome inhibitor). Nasal basal cells show a similar behavior to keratinocytes with a partial inhibition of cell death induced by Portimine and Anisomycin whereas endothelial cells are insensitive to bortezomib. This suggests to us that, as described in this study and in the newly generated results presented in point 2 but also in previous studies, ZAK α has a broader role than the sole activation of NLRP1 and that this pathway can also lead to additional cell death processes such as apoptosis or pyroptosis driven by Caspase-3 or additional Caspases. In addition, NLRP1, although being described to be expressed in some endothelial cells, might also be regulated differently in those cells. Such observation has also been done by others in naïve T cells for instance, where those cells do not engage NLRP1 upon VbP exposure, but rather CARD8 (Linder *et al*, 2020; Johnson *et al*, 2020).
- Regarding the experiments on monocytes, we exposed human blood monocytes or keratinocytes to Portimine and anisomycin. Analysis of cell death showed that anisomycin but not Portimine triggered monocyte death. This was also confirmed by the use of the ZAKalpha inhibitor PLX4720 and by addressing the level of P38 phosphorylation (activated by ZAKalpha upon RSR induction). Only anisomycin could promote P38 phosphorylation in monocytes. This constitutes to us an intriguing point that we are currently investigating but for which we do not have yet any proper answer.

These results have been included in the revised version of the manuscript and are presented in **Fig. EV3**.

A**B**
Figure: A, B. Phosphotag blotting of phosphorylated P38 and cell lysis (LDH) evaluation in pHEKs, endothelial cells, nasal basal cells and human blood monocytes after 24 hours exposure to pure Portimine A (4ng/mL) or Anismoycin (1 μ g/mL and 10 μ g/mL in monocytes). When specified the compounds PLX4720 (ZAK α , 10 μ M), Emericasan (pan Caspase inhibitor, 5 μ M) and bortezomib (proteasome inhibitor, 1 μ M) were used. ***P \leq 0.001, two-way ANOVA with multiple comparisons. Values are expressed as mean \pm SEM. Graphs show one experiment performed in triplicates at least three times.

2. The authors clearly show that Portimine A, like anisomycin, induces some non-NLRP1 dependent cell death (Fig. 1H). As such, they should include a blot for cleaved N-terminal GSDMD and a marker of apoptosis (PARP, cleaved casp-3) in these experiments.

Indeed, this has been an intriguing point to us as well. Accordingly, we performed additional experiments and observed that Gasdermin D active fragment (p30) was well induced in WT keratinocytes exposed to Portimine A and Anisomycin (positive control). In addition, this fragment disappeared in NLRP1-deficient keratinocytes, supporting the importance of NLRP1 at promoting subsequent Caspase-1-dependent GSDMD cleavage upon Portimine A exposure. Yet, we also observed the presence of the Caspase-3-generated GSDMD p43 and GSDME p35 fragments in response to both Portimine and Anisomycin. This was further supported by the presence of active Caspase-3 (cleaved fragments) in those conditions. Furthermore, such process was found to be NLRP1-independent, hence suggesting that Caspase-3 might also be involved in this NLRP1-independent cell death induced by RSRs, including Portimine. In this context, inhibiting Caspase-3 in NLRP1-deficient keratinocytes entirely abrogated Portimine-induced cell death, confirming that this second cell death pathway is also present upon Portimine exposure. At this step, it is too early to scientifically state about Caspase-3 importance in Portimine-induced skin pathology but, as pointed by our reviewer in **points 1, 2 and 5** and discussed below, the importance of ZAKalpha in the zebrafish pathology is more prominent than NLRP1 and Asc, which suggests that this caspase-3 pathway might also be contributing to Portimine-driven pathology. To this regard, this is a point we are addressing in details in the frame of a broader study.

The results generated are now included in the **Fig. 2** of the revised manuscript.

lysis (LDH) and IL-1β release evaluation in pHEKs and NLRP1-deficient pHEKs upon pure Portimine A (4ng/mL) or Anisomycin (1μg/mL) exposure for 24 h. When specified, the Caspase-3 inhibitor (Z-DEVD, 20 μM) was used. ***P ≤ 0.001, two-way ANOVA with multiple comparisons. Values are expressed as mean ± SEM. Graphs show one experiment performed in triplicates at least three times.

Figure: A. Immunoblotting of GSDMD, GSDME, Caspase-3, NLRP1 (C-term part) and tubulin in pHEKs and NLRP1-deficient pHEKs (generated in B) after 24 hours exposure to Portimine A (4ng/mL) or to the known RSR inducer Anisomycin (1μg/mL). Immunoblots show lysates from one experiment performed at least two times. B. Cell

3. They identify several NLRP1 mutations that confer resistance to portimine. They should confirm that NLRP1 function is still intact by showing responsiveness to Val-boroPro in these lines.

In agreement with our reviewer comment, we included our VbP controls in our figure. As shown below, they show that Vbp (10 μ M, 12 hours) still hold the capacity to promote NLRP1 response, independently of the variants studied, contrary to Portimine A. This strongly suggests that those variants are affecting the pathway of phosphorylation-driven NLRP1 activation but not the DPP8/9 regulatory pathway inhibited by VbP. The figure is now updated in the revised version of the study and referenced in the **Fig. 3** panel.

4. The authors should perform genome mining for marine species for NLRP1 (or related) proteins that contain the ZAKa motif, so that they can speculate on the species (in marine environments) that may respond to Portimine A. Especially as this motif is not conserved in mice.

This is indeed a very important point that we are currently addressing. We would like to discuss about several things here:

- First, the specific molecular target of Portimine A still remains controversial as a recent study showed that Portimine A directly binds to NMD3, an essential cytosolic protein required for the assembly of the cytosolic 60S ribosome (Tang *et al*, 2023). Curiously, knocking down NMD3 confers protection against Portimine A in cancer cell lines, rather than eliciting the same effects as Portimine A (Tang *et al*, 2023). These results suggest that Portimine A is unlikely to act directly as an inhibitor of NMD3, which raises the key question of the main target of Portimine A in eukaryotic species. This is currently under strong investigations.
- Second, although ZAKalpha and NLRP1 are major inducers of inflammatory responses in human and zebrafish, they are not the main targets of Portimine A. In theory, a specie that would lack a conservation of the NLRP1 or ZAKalpha pathways with humans but where Portimine A would still target ribosomes, would still respond to Portimine A. This has been nicely shown recently by two independent groups with UVB and Anisomycin driving RSR-dependent pathologies both in human and in rodent models, although rodent do not connect ZAKalpha to NLRP1 (Sinha *et al*, 2024; Vind *et al*, 2024). This strongly suggests that NLRP1 has a specific function in specific species, including humans, but that the direct target that leads to ribosome inactivation might be the most promising way to investigate which marine species are mainly targeted by Portimine A.

Overall, although we agree with our reviewer that searching for common and different determinants of Portimine A sensitivity in various marine species is of strong interest, which we are currently addressing by structural analyses, we believe that at this step it is too preliminary and goes beyond the frame of this study. In this context, we respectfully ask not to include those genome mining search as it is too preliminary to speculate in broader scale than the human level in absence of identification of the direct target of Portimine A.

5. The data that NLRP1 is responsible for Portimine pathology in zebrafish is weak (only small changes seen), and it is not clear how statistically significant this is. Far more data, including evidence of pyroptosis (GSDMD cleavage), is needed to confirm that this natural product really activates the NLRP1 inflammasome in these fish.

We agree that the NLRP1 contribution in zebrafish is lower than the ZAKa contribution in response to Portimine A. This is more discussed in the revised version of the manuscript and supported by the fact that NLRP1 is one branch activated by the ZAKalpha pathway and that additional pathways triggered by ZAKalpha might also contribute to Portimine A-driven pathology, such as for instance Caspase-3 activation and cell death.

Regarding the zebrafish model of NLRP1 response, we performed additional experiments. First, in zebrafishes NLRP1 form an inflammasome through its adaptor Asc (Li *et al*, 2018) like in humans. Therefore, we genetically invalidated Asc in zebrafish in addition to NLRP1 and ZAKa. Our new results, where we determined the impact of Portimine A on fin tail size (area) and damage showed that zebrafishes lacking ZAKa had a strong decrease in fin tail loss of area and in the accumulation of skin damages. We also observed that strains lacking Asc exhibited a close phenotype to NLRP1 deficient larvae in response to Portimine A, yet less pronounced than in the ZAKa-deficient strains. This suggests to us two important points: Zebrafish ZAKa seems to respond in a similar way than in human cells to Portimine A exposure and the zebrafish NLRP1-Asc pathway also responds to Portimine A but exhibits a lower contribution than ZAKa, similarly to what we observed in human cells.

Gasdermin D is not conserved in zebrafish that rather express two GSDME isoforms (Gsdmea and Gsdmeb) that can be cleaved by two caspase proteases (caspase a and caspase b). However, previous studies from the Leptin lab clearly showed that none of those two Gasdermins (Gsdmea and Gsdmeb) was involved in zebrafish inflammasome-driven keratinocyte death, hence refraining the authors and ourself calling this death “pyroptosis” at this step (Hasel de Carvalho *et al*, 2023). In this context, we took advantage of previous studies that demonstrated that inflammasome in zebrafish could also mediate neutrophil recruitment (Rodríguez-Ruiz *et al*, 2023). By generating ZAKa and NLRP1 ko in the fluorescent neutrophil line LysC::GFP we could monitor for neutrophil recruitment in the caudal fin tail and the respective contribution of ZAKa and NLRP1 pathways. We could observe that Portimine A induced significant recruitment of neutrophils in the caudal fin, suggesting that an inflammatory process was at play. Importantly, ZAKa and NLRP1 deficiency strongly dampened neutrophil recruitment, hence confirming the importance of ZAKa in this process but also validating that the zebrafish NLRP1 also exhibits a significant contribution in this process during Portimine A exposure.

Figure: A. Determination and quantification of zebrafish larvae fin tail damage induced by Portimine A (75nM) in WT larvae (20/group). Two specific parameters were studied for damages quantifications, namely the Tail area ($\times 10^5 \mu\text{m}^2$) and the % of damage severity. **B.** CRISPR/Cas9 gRNA strategy used to genetically ablate ZAK α , Asc and NLRP1 in zebrafish embryo (for details see material and methods). **C.** Determination and quantification of zebrafish larvae fin tail damage induced by Portimine A (75nM) in WT, Nlrp1-, Asc- and ZAK α -deficient larvae (20/group) after 30 hours. **** $P \leq 0.0001$, one-way ANOVA (Kruskal-Wallis test). **D.** Determination and quantification of zebrafish neutrophil recruitment to the fin tail in WT and Nlrp1- and ZAK α -deficient larvae (20/group) of the zebrafish line Tg(LysC:GFP)^{nz117} with GFP-expressing neutrophils over 24 hours exposure to Portimine A (75ng/mL). Graph shows a representative experiment out of the two performed. Values are expressed as mean. **** $P \leq 0.0001$, one-way ANOVA (Kruskal-Wallis test). NS. Not significant.

6. While the authors show that portimine a is in the marine samples, and that portimine a can activate human NLRP1, they do not definitively show that protamine A is indeed responsible for the skin disease. The discussion should make clear that this was not unequivocally shown.

This is indeed an important point. We agree that at this step, the lack of patient skin biopsies and clear quantification of Portimine A presence in those biopsies but also the determination in those samples of NLRP1/ZAK activation is missing to definitely state that Portimine A is the causative agent of the observed dermatitis in fishermen of Senegal and hobbyists in Cuba. Despite our efforts, we did not obtain those samples from patients. In this context, we added a paragraph at the end of the discussion in order to state that at this step, we cannot formally exclude additional causes/inducers might also cause this dermatitis in the fishermen of Senegal and hobbyists in Cuba.

“Finally, although our study does not provide the final evidence that during the outbreaks in Cuba and in Senegal, Portimine A was directly responsible of the strong dermatitis observed, our results strongly point toward this direction and warrant for the necessity to address *V. rugosum* presence and amounts in various sea areas in order to prevent any adverse effects on the human health.”

Other:

Fig3A, typo: "liquide"

This has been corrected in the figure panel.

References

- Hasel de Carvalho E, Dharmadhikari SS, Shkarina K, Xiong JR, Reversade B, Broz P & Leptin M (2023) The Opto-inflammasome in zebrafish as a tool to study cell and tissue responses to speck formation and cell death. *Elife* 12
- Johnson DC, Okondo MC, Orth EL, Rao SD, Huang HC, Ball DP & Bachovchin DA (2020) DPP8/9 inhibitors activate the CARD8 inflammasome in resting lymphocytes. *Cell Death Dis* 2020 118 11: 1–10
- Li J, Gao K, Shao T, Fan D, Hu C, Sun C, Dong W, Lin A, Xiang L & Shao J (2018) Characterization of an NLRP1 Inflammasome from Zebrafish Reveals a Unique Sequential Activation Mechanism Underlying Inflammatory Caspases in Ancient Vertebrates. *J Immunol* 201: 1946–1966
- Linder A, Bauernfried S, Cheng Y, Albanese M, Jung C, Keppler OT & Hornung V (2020) CARD8 inflammasome activation triggers pyroptosis in human T cells. *EMBO J* 39
- Rodríguez-Ruiz L, Lozano-Gil JM, Naranjo-Sánchez E, Martínez-Balsalobre E, Martínez-López A, Lachaud C, Blanquer M, Phung TK, García-Moreno D, Cayuela ML, *et al* (2023) ZAK α / P38 kinase signaling pathway regulates hematopoiesis by activating the NLRP1 inflammasome . *EMBO Mol Med* 15
- Sinha NK, McKenney C, Yeow ZY, Li JJ, Nam KH, Yaron-Barir TM, Johnson JL, Huntsman EM, Cantley LC, Ordureau A, *et al* (2024) The ribotoxic stress response drives UV-mediated cell death. *Cell* 187: 3652-3670.e40
- Tang J, Li W, Chiu TY, Martínez-Peña F, Luo Z, Chong CT, Wei Q, Gazaniga N, West TJ, See YY, *et al* (2023) Synthesis of portimines reveals the basis of their anti-cancer activity. *Nature* 622: 507–513
- Vind AC, Wu Z, Firdaus MJ, Snieckute G, Toh GA, Jessen M, Martínez JF, Haahr P, Andersen TL, Blasius M, *et al* (2024) The ribotoxic stress response drives acute inflammation, cell death, and epidermal thickening in UV-irradiated skin in vivo. *Mol Cell* 0

15th Jan 2025

Dear Dr. Meunier,

Thank you for submitting your revised study, and please accept my apologies for the delay in getting back to you during this busy time of the year. We have now received the reports from referees #2 and #3, who also evaluated your responses to referee #1.

As you will see from the reports below, they are mostly satisfied with the revisions, and I will therefore be able to accept your manuscript once the following issues are addressed:

1/ Referees' comments:

Please address the remaining concern from referee #3.

2/ Manuscript text:

- Please remove the highlights in the text, and only keep in track changes mode any new modification.
- The email addressed to Thomas.besnoit@ipbs.fr bounced, please provide a corrected address.
- Please provide up to 5 keywords.
- Kindly correct the order of the manuscript sections as follows: Abstract, Keywords, The Paper Explained, Introduction, Results, Discussion, Methods, Data Availability, Acknowledgements, Disclosure and competing interests statement, References, Figure legends, Tables and their legends, Expanded View Figure legends.
- Methods:
 - o Please remove the Material Availability section.
 - o Please download and fill our Reagents and Tools Table template (.docx), which you can find in our author guidelines: <https://www.embopress.org/page/journal/14693178/authorguide#structuredmethods>. Do not include the Reagents and Tools Table in the Methods section of the manuscript but upload it as a separate file choosing the file type "Reagent Table" (i.e. remove current Table 1 from the manuscript text).
 - o Patient material: please include a statement confirming that informed consent was obtained from all subjects and that the experiments conformed to the principles set out in the WMA Declaration of Helsinki and the Department of Health and Human Services Belmont Report.
 - o Cell material: Please indicate whether the cells were authenticated and tested for mycoplasma contamination.
 - o Statistical analysis: please provide a statement on exclusion/inclusion criteria, blinding and randomization.
- Acknowledgements: The information provided in the manuscript and the submission system should match, please adjust accordingly.
- Please rename 'Conflict of interest' to 'Disclosure statement and competing interests'. Please review our updated policy <https://www.embopress.org/competing-interests> and update your competing interests if necessary.

3/ Figures:

- Figure EV3B NMD3: There is a conversion line through the blot. Please re-convert this figure to remove the line.
- The panel "a" in Figure EV should be removed since it's the only panel.
- The tables in the manuscript text should get titles and callouts, and be moved after the main figure legends.
- Please address the queries from our copy editors in the figure legends:
 1. Please define the annotated p values ****/**/*/* as well as provide the exact p-values for the same in the legend of figure 4F as appropriate..
 2. Please note that the exact p values are not provided in the legends of figures 1E, F; 2A, B, C, E; 3D, E, G; 4B, C, E, G; EV1 E, F, G; EV3 E, F; EV4 D.
 3. Please indicate what */ **/ ***/ **** represents; if this represents p value(s), please indicate the statistical test and the exact p value in the legend(s) of figure(s) 4D; EV4 B.
 4. Please note that information related to n is missing in the legends of figures 4E, G; EV4 B
 5. Please note that the scale bar needs to be defined for figures 4B.

4/ Checklist:

- Experimental study design and statistics: please fill in all fields of this section.
- Ethics: please fill in the section "Studies involving human participants, informed consent and Helsinki declaration"
- Ethics: please fill in the section "Studies involving specimen and field samples"

5/ Please note that all corresponding authors are required to supply an ORCID ID for their name upon submission of a revised manuscript. An ORCID identifier is currently missing for P. Brehmer, F. Roca and J. Common.

6/ Synopsis:

- Please reformat your synopsis image to 550 px wide x 300-600 px high. A cropped portion of this image will serve as thumbnail for the table of content on our webpage.

- I slightly edited your synopsis text, please let me know if you agree with the following or amend as you see fit:
"In 2020-2021, a mysterious skin disease affected fishermen in Senegal and Cuba.
- High levels of Portimine A toxin, produced by the dinoflagellate *Vulcanodinium rugosum*, were found in Senegal's fishing zones.
- Portimine A induced a strong inflammatory response in human skin epithelial cells.
- Portimine A-inhibited translation supported a ribotoxic stress response mediated by ZAK α kinase.
- ZAK α promoted NLRP1 inflammasome-dependent skin inflammation in response to Portimine A in human skin models and in zebrafish models."

7/ As part of the EMBO Publications transparent editorial process initiative (see our Editorial at <http://embomolmed.embopress.org/content/2/9/329>), EMBO Molecular Medicine will publish online a Review Process File (RPF) to accompany accepted manuscripts.

This file will be published in conjunction with your paper and will include the anonymous referee reports, your point-by-point response and all pertinent correspondence relating to the manuscript. Let us know whether you agree with the publication of the RPF and as here, if you want to remove or not any figures from it prior to publication.

I look forward to receiving your revised manuscript.

Yours sincerely,

Lise Roth

***** Reviewer's comments *****

Referee #2 (Remarks for Author):

The authors have made significant improvements to the manuscript, addressing most of the reviewers' concerns. The revised figure clearly demonstrates that Portimine A induces phosphorylation of NLRP1 via phosphotag analysis, greatly enhancing clarity. Additionally, the new data confirm that monocytes have an intact ZAK α pathway and show that Anisomycin, but not Portimine A, triggers significant monocyte death. These revisions strengthen the manuscript and provide valuable insights into the underlying mechanisms.

Referee #3 (Comments on Novelty/Model System for Author):

This manuscript still has a major issue in it the link between portamine and skin disease is correlative, not causative. The title and abstract clearly state that they have revealed the illness comes from the toxin, but this is not the case. This should be edited for scientific accuracy.

Referee #3 (Remarks for Author):

The authors still claim (especially in abstract) that portamine causes the skin disease, but they have not shown this to be the case. The text should be edited for accuracy.

***** Reviewer's comments *****

We thank our reviewers for their comments, suggestions and critics. Allover they strongly improved the scientific findings of this study. We have, according to the last reviewer 2 suggestions, modified our title and abstract for better scientific accuracy.

Referee #2 (Remarks for Author):

The authors have made significant improvements to the manuscript, addressing most of the reviewers' concerns. The revised figure clearly demonstrates that Portimine A induces phosphorylation of NLRP1 via phosphotag analysis, greatly enhancing clarity. Additionally, the new data confirm that monocytes have an intact ZAK α pathway and show that Anisomycin, but not Portimine A, triggers significant monocyte death. These revisions strengthen the manuscript and provide valuable insights into the underlying mechanisms.

Referee #3 (Comments on Novelty/Model System for Author):

This manuscript still has a major issue in it the link between portamine and skin disease is correlative, not causative. The title and abstract clearly state that they have revealed the illness comes from the toxin, but this is not the case. This should be edited for scientific accuracy.

Referee #3 (Remarks for Author):

The authors still claim (especially in abstract) that portamine causes the skin disease, but they have not shown this to be the case. The text should be edited for accuracy.

Dear Lise,

Thank you for your email.

Regarding the point 1, yes I confirm that each one of the 6 co-last author takes full responsibility for the study. I know this is unusual. But, in this study, we had to face a very transversal expertise and coordination that involved A/ P. Brehmer and P. Hess work on the field (in Senegal, with working with local authorities, convincing the fishermen and authorities to get access to samples, to collect the samples, to characterize the samples), B/ FL Zhong/myself who coordinated the immune detection on the inflammatory response in keratinocytes and identifying the molecular mechanisms involved and C/ F. Roca/J Commons who coordinated the whole pathophysiological models of exposure. Each expertise is unique and complementary and fully justify this "unusual" aspect of sharing altogether the corresponding authorship, including as well the responsibility for the whole study. I hope it is more clear now.

2/ For the picture, attached are different versions where we tried improving the readability of them. Tell me if this is ok with you.

Regards and thanks you for your help

Etienne

23rd Jan 2025

Dear Dr. Meunier,

Thank you for submitting your revised files. I am pleased to inform you that your manuscript is accepted for publication and is now being sent to our publisher to be included in the next available issue of EMBO Molecular Medicine!

If you have any questions, please do not hesitate to contact the Editorial Office.

Thank you for your contribution to EMBO Molecular Medicine.

With kind regards,

Lise Roth
